# Structural insights into intron catalysis and dynamics during splicing

Ling Xu[1,2,5 ✉], Tianshuo Liu[2,5], Kevin Chung[3,5] & Anna Marie Pyle[1,2,4 ✉]

The group II intron ribonucleoprotein is an archetypal splicing system with numerous mechanistic parallels to the spliceosome, including excision of lariat introns[1,2]. Despite the importance of branching in RNA metabolism, structural understanding of this process has remained elusive. Here we present a comprehensive analysis of three single-particle cryogenic electron microscopy structures captured along the splicing pathway. They reveal the network of molecular interactions that specifies the branchpoint adenosine and positions key functional groups to catalyse lariat formation and coordinate exon ligation. The structures also reveal conformational rearrangements of the branch helix and the mechanism of splice site exchange that facilitate the transition from branching to ligation. These findings shed light on the evolution of splicing and highlight the conservation of structural components, catalytic mechanism and dynamical strategies retained through time in premessenger RNA splicing machines.

Splicing lies at the heart of RNA metabolism in eukaryotes. During this indispensable stage of gene expression, introns are removed from premessenger RNA transcripts to generate mature messenger RNAs (mRNAs)[1,3,4] (Fig. 1a). The modern spliceosome, the molecular machine that executes the splicing reaction, is thought to originate from the same ancestral molecule as the self-splicing group II introns that are still commonly found in bacteria and organelles of plants and fungi[2]. Group II introns are large ribozymes that catalyse their own excision from precursor RNA transcripts[5]. Both splicing machineries form a conserved active site that hosts the catalytically essential heteronuclear metal ion core[6,7]. Moreover, they both branch using a bulged adenosine nucleophile, forming the distinctive lariat intron featuring a 2′,5′-linked phosphodiester linkage. Intron D4 contains an open reading frame (ORF) that encodes a specialized multidomain protein (the 'maturase', Fig. 1b,c) which shares strong structural similarity to Prp8, a central protein component of the U5 snRNP[8,9]. Through formation of a ribonucleoprotein (RNP) holoenzyme with the parent intron RNA, the maturase facilitates intron splicing out of the transcript as well as retrohoming into new genomic loci[10].

In light of these structural and mechanistic similarities, group II intron RNPs have become a prototypical system for studying the general biochemical principles of RNA splicing and the molecular evolution of splicing machines[11]. Despite advances in the visualization of group II intron RNAs[7,12–14] and RNPs[15–17], the structural organization of group II intron systems as they undergo branching and then coordinate the two steps of splicing has remained elusive. Therefore, it is unclear how group II introns properly recognize the branchpoint and the 5′-splice site (5′SS) and how maturases facilitate the branching reaction. These questions are of vital importance as they provide clues on the origin of intron branching, which is among the most fundamental reactions in RNA biology.

To visualize the conformational states along the group II intron RNP splicing pathway, we chose the group IIC intron from *Eubacterium*

*rectale* and its encoded maturase, MarathonRT[18], as the model system. The maturase acts as a branching switch that shifts the intron splicing pathway from hydrolysis to branching (Extended Data Fig. 1a–d). Here, we used single-particle cryogenic electron microscopy (cryoEM) to obtain the structures of the RNP at each sequential stage during splicing. By capturing the RNP in the state immediately before branching (3.0 Å overall), we visualized how the branchpoint adenosine (bpA) and the splice site (SS) are held in place through molecular interactions between the branch helix and conserved regions of the RNP. Our structural observations reveal a close resemblance between group II intron RNPs and the spliceosome in terms of branchpoint recognition and branch helix positioning. We also gained unique insights into the strategy by which the attacking 2′-OH is precisely positioned in activation distance to the catalytic metal M1, ready for nucleophilic attack. This high-resolution view enables us to construct a complete and catalytically relevant molecular picture of the splicing active site before branching, which has largely eluded structural characterization despite earlier hints[19].

In addition to the prebranching RNP structure, we present the RNP structures preceding and following the exon ligation step. These structures allow us to visualize large movements of the branch helix, local movements of the branchpoint and the SS exchange that occurs between the two steps of splicing. The conformational dynamics of the spliceosome branch helix recapitulates that of the group II intron, thereby demonstrating that branch helix dynamics are a conserved physical attribute that is codified in splicing machines.

## Capturing group II intron RNP in action

To elucidate the mechanism of forward splicing through branching, we sought to visualize structures of the group II RNP complex at each stage along the branching pathway. The two chemical steps of group

[1]Howard Hughes Medical Institute, Chevy Chase, MD, USA. [2]Department of Molecular, Cellular and Developmental Biology, Yale University, New Haven, CT, USA. [3]Department of Molecular Biophysics and Biochemistry, Yale University, New Haven, CT, USA. [4]Department of Chemistry, Yale University, New Haven, CT, USA. [5]These authors contributed equally: Ling Xu, Tianshuo Liu, Kevin Chung. ✉e-mail: ling.xu@yale.edu; anna.pyle@yale.edu

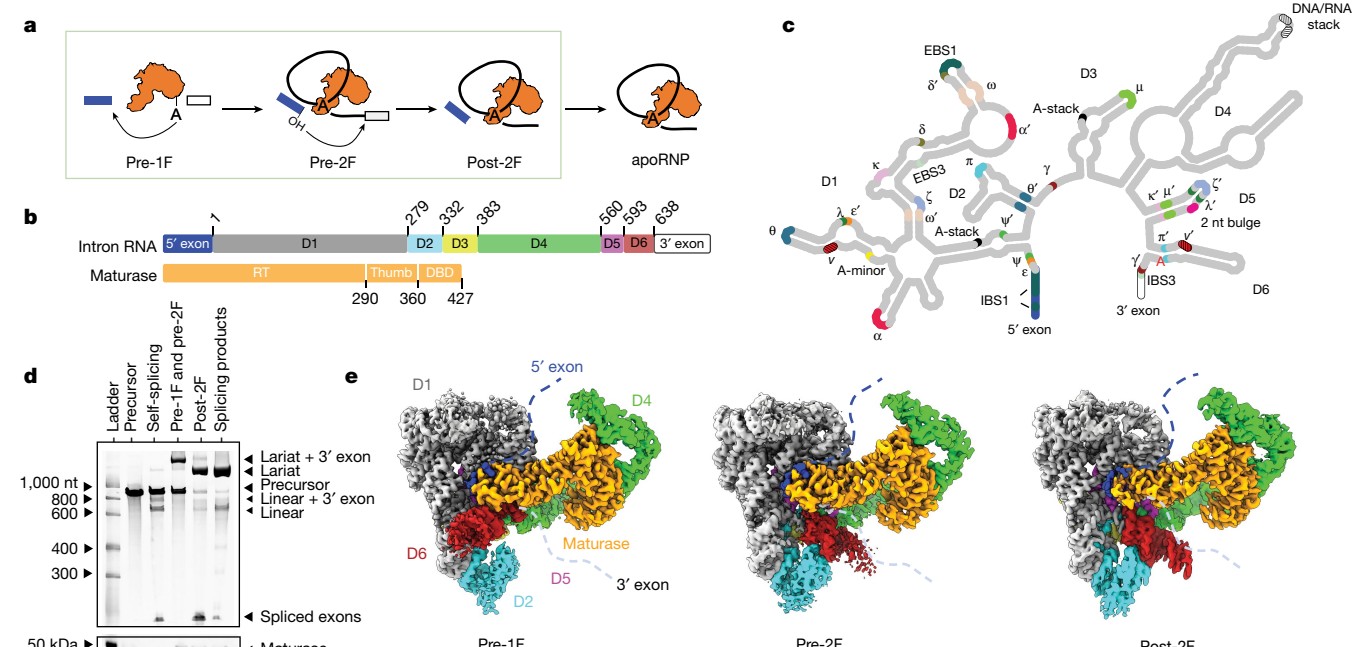

**Fig. 1 | CryoEM reconstructions of a group IIC RNP undergoing the branching reaction. a**, Cartoon of group II RNP splicing. **b**, Domain organization of the intron RNA and its maturase. **c**, Secondary structure of the intron, with annotated tertiary interactions. **d**, A GelRed-stained 5% urea–polyacrylamide gel electrophoresis (PAGE) gel (top) and a SYPRO Ruby-stained SDS–PAGE gel (bottom) showing various conditions used to obtain samples for cryoEM. Lane 1 is the size marker for RNA (top) and protein (bottom). Lane 2 is the marker showing migration of the precursor RNA. Lanes 3 and 6 are the reaction ladders showing migration of the linear and lariat products, respectively. Lanes 4 and 5 are independent cryoEM samples captured at various reaction stages. **e**, Composite cryoEM maps of the prebranching (pre-1F), preligation (pre-2F) and postligation (post-2F) RNP complexes.

II intron splicing are spontaneous and do not require energy input or step-specific factors. It is therefore challenging to stall the reaction without disrupting the active site and earlier attempts caused conformational distortions that made it difficult to discern the exact molecular mechanism of branching[20]. To resolve this, we incubated splicing precursor constructs in the presence of maturase protein, replacing $Mg^{2+}$ with $Ca^{2+}$ to yield complexes stalled in the precursor and branching intermediate states (Fig. 1d and Extended Data Fig. 1e,f). To obtain the postligation RNP, we assembled the lariat apoRNP[17] with an oligonucleotide equivalent to the ligated exon, thereby enabling us to investigate the structural changes upon completion of intron splicing (Fig. 1d).

The corresponding RNP samples were vitrified on grids and appeared as monodispersed particles on cryoEM micrographs, suitable for structure determination. As these samples show preferred orientation, we used the Chameleon system (Spotiton)[21,22] and combined this with tilted datasets[23] to obtain a uniform angular distribution (Extended Data Figs. 2 and 3). The increased diversity of particle orientations allowed us to generate two distinct, isotropic maps. Upon inspection of the two reconstructions, we assigned the corresponding maps to the prebranching (pre-1F) and preligation (pre-2F) states, respectively (Fig. 1e). The resolution of these maps approaches 2.8 and 2.9 Å, respectively, for the catalytic core (Extended Data Fig. 3). We obtained a three-dimensional (3D) reconstruction for the postligation RNP (post-2F) and the resolution of the catalytic core was determined to 2.9 Å (Fig. 1e and Extended Data Fig. 4). This collection of structures allows us to present a full molecular picture of the group II intron RNP as it proceeds along the branching pathway.

## Positioning of the branch helix

To splice through the branching pathway, the group II RNP forms intramolecular RNA and intermolecular RNA–protein interactions that precisely arrange the branch helix (D6) in the branching-competent conformation (Fig. 2a and Supplementary Video 1). The intron scaffold domain, D1, contributes to this by forming an extended, interlocked interaction network between D1c and D6 (denoted ν–ν′) (Fig. 2b). This network features a long-range base pair between G86 and C601, both of which are bulged nucleotides with strong conservation signatures (Fig. 2c and Extended Data Fig. 5a,b). Consistent with their significance in D6 positioning, deletion of G86, C601 or both, leads to branching defects, whereas substituting this GC pair for an AU pair partially rescues branching (Fig. 2d). A wobble pair between G84 and U104 in D1c anchors another intricate molecular network around C601 to further restrain the conformational sampling of D6. In agreement with our structural observations, a G84A/G86A dual mutation, which does not alter D1c secondary structure, has the most pronounced deleterious effect (Fig. 2d). Hence, our results highlight the active role of interdomain RNA interactions in proper positioning of D6.

On the opposite side of the D6 helix, the thumb and DBD domains of the maturase protein compose another extensive RNA–protein intermolecular interface, where we visualized three clusters of interactions. The first cluster (Trp310, Ser313 and Gln359) is located at the basal stem of D6 (Fig. 2e) and its disruption, through mutation to alanine, abolishes intron branching (Fig. 2f). At the junction loop between the thumb and DBD domain a second cluster of residues (Thr362 and Asn365) grasps the central section of D6 adjacent to the branch site and the ribozyme active site. In addition to phosphate backbone interactions, a highly conserved lysine residue (Lys361) inserts into the main groove of D6 and makes direct contact with the 5′SS (G1:N7), juxtaposing the first-step nucleophile with the scissile phosphate (Fig. 2e and Extended Data Fig. 5c). Consistent with this structural observation, a Lys361 single alanine mutant is sufficient to eliminate branching activity, as does the Lys361/Thr362/Asn365 triple mutant (Fig. 2f). Finally, the side chains of Lys372 and Arg377 in the DBD domain grip the distal, upper stem of D6 and mutations introduced at these sites compromise branching.

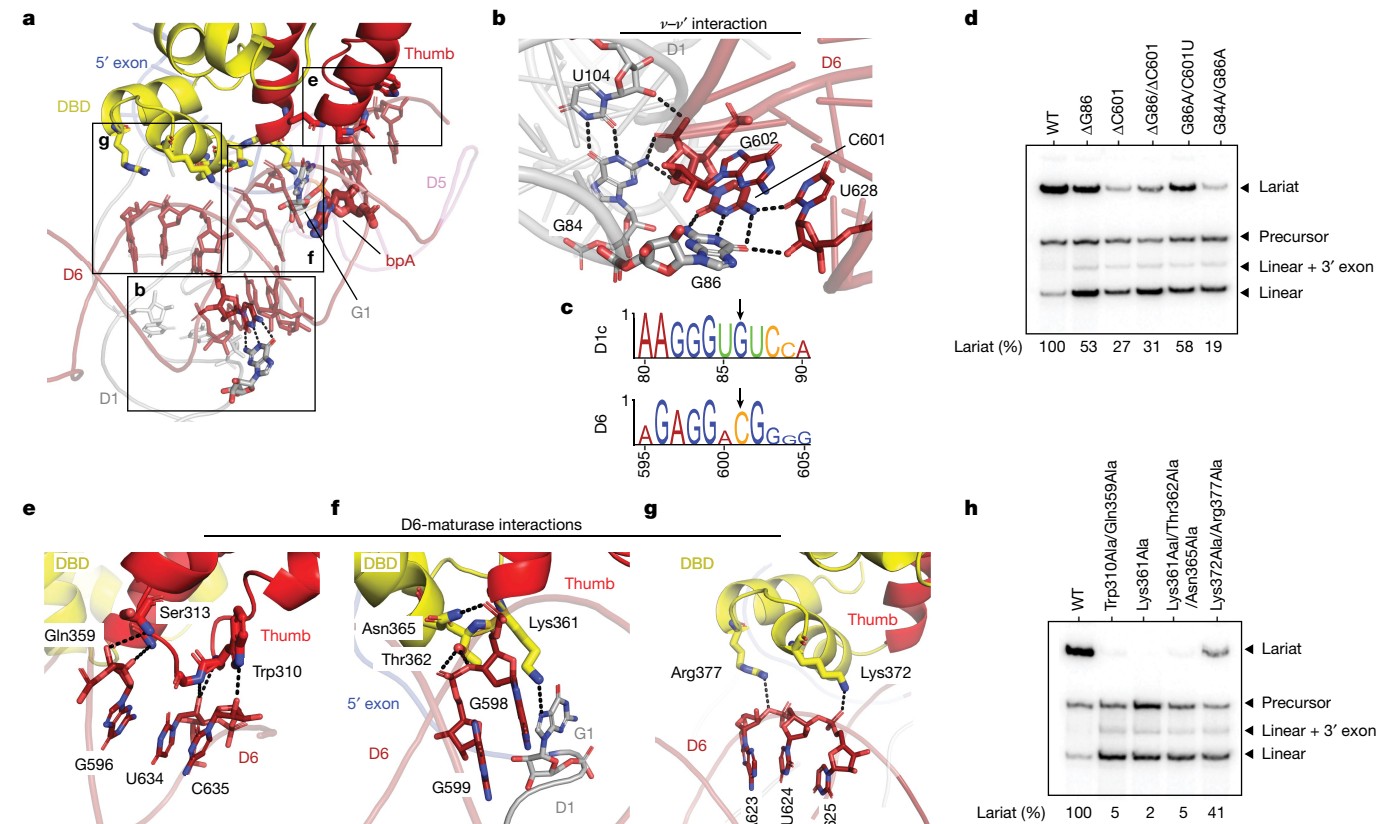

**Fig. 2 | Interactions of D6 with the intron RNP before lariat formation.**
**a**, Interaction network surrounding the D6 helix before branching (boxed elements are labelled with the figure panel designations described next). **b**, Newly discovered long-range RNA interaction (*ν–ν′*) that positions D6. **c**, Conservation of nucleotides involved in **b**. **d**, A denaturing radioanalytical splicing gel showing effects of intron mutants in the presence of WT maturase protein. Individual data points representative of *n* = 4 in vitro splicing assays are

shown. **e**, Intron–maturase interaction cluster at the basal region of the D6 helix. **f**, Intron–maturase interaction cluster at the central region of the D6 helix. **g**, Intron–maturase interaction cluster at the distal region of the D6 helix. **h**, A denaturing radioanalytical splicing gel demonstrating effects of maturase mutants on promoting branching of WT intron construct. Individual data points representative of *n* = 4 in vitro splicing assays are shown.

Using this vast molecular network, the maturase protein stabilizes the branching-competent conformation of D6 and brings the 5′SS adjacent to the branchpoint, thereby explaining why it is indispensable for promoting branching (Extended Data Fig. 1).

In the D6-docked state, one observes an interplay between the intron 5′SS and the branch helix, where the first two nucleotides expose their Watson–Crick edges to engage in tertiary interactions with D6. Specifically, G1:O6 engages the 2′-OH of C633, the nucleotide next to the branchpoint, which secures D6 and brings the branchpoint close to the 5′SS. U2 further strengthens contacts between the 5′SS and D6 through a base triple interaction with the G599-C629 base pair (Extended Data Fig. 6a). The pre-1F structure therefore provides a glimpse into the group II intron 5′SS and establishes its role in branch helix positioning, explaining the conservation signature of the group II intron 5′SS (ref. 24).

## Branchpoint recognition and dynamics

Having elucidated the mechanism by which the branch helix is docked in the prebranching conformation, we next sought to unveil the branchpoint recognition strategy. This question is of vital importance, as stringent branchpoint use is a hallmark of group II intron splicing[25] and yet the interaction network that recognizes and activates the branchpoint nucleotide for catalysis has eluded structural characterization.

Our prebranching map, with a local resolution of 2.8 Å around the branchpoint, allows confident model building which reveals the

structural basis of branch site recognition. We show that the bpA (A632) is recognized by means of a base triple interaction with the G598-C630 base pair. The exocyclic amine of the bpA (bpA:N6) forms a crucial hydrogen bond with O2 of C630 (Fig. 3a), consistent with previous chemical genetics studies[25]. The interaction partner (C630) is located two nucleotides upstream and, based on covariation analysis of group II introns[26], it is almost exclusively a pyrimidine. The 2′-OH of C630 forms an extra hydrogen bond with bpA:N1. This interaction serves as an extra molecular lock to hold the branch site in place. Intriguingly, the molecular recognition pattern we observe in group II introns is identical to that reported in structures of the spliceosome[6,27]. Recent models for the yeast C complex[6] (postbranching) revealed the same molecular interaction between the bpA and a highly conserved uridine located two nucleotides upstream (Fig. 3b), which can reasonably fit in the density of the yeast B* complex (prebranching). A similar bpA recognition strategy has also been proposed in the C complex of the human spliceosome[27]. Our structure therefore unveils a mechanistic parallel in splicing machines for defining the branchpoint, which seems to be hard-coded by molecular evolution.

Next, we sought to visualize the local conformational dynamics of the bpA along the branching pathway. In the pre-1F state, the bpA adopts an unusual conformation that causes it to point toward the main groove of the branch helix, through a base triple interaction. This conformation leads to significant distortion of the bpA sugar–phosphate backbone (Fig. 3a,d), which places the 2′-OH next to the scissile phosphate. After branching, as the complex enters the preligation state, the bpA flips to

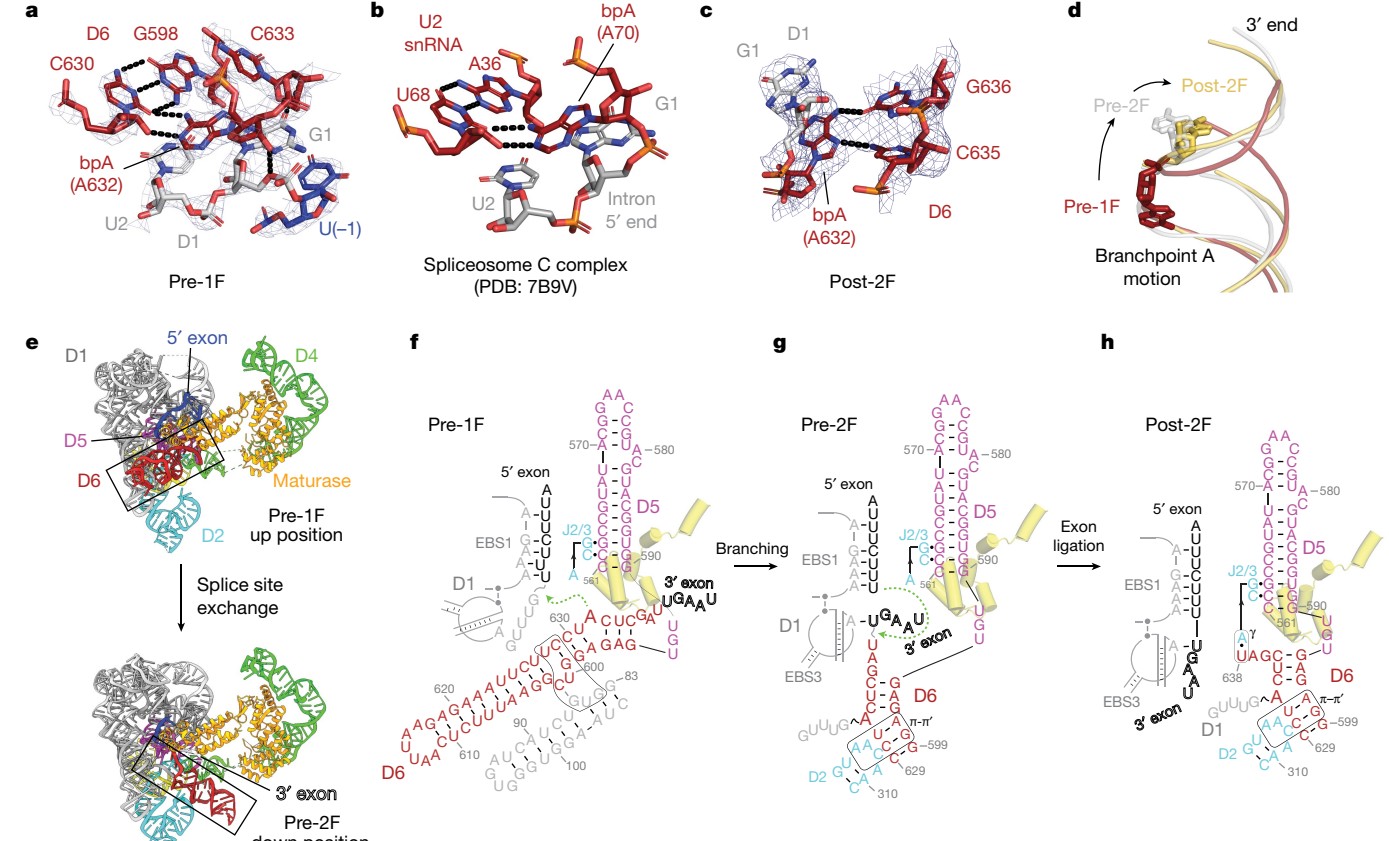

**Fig. 3 | Molecular recognition of the branchpoint A and D6 dynamics.**
**a**, Interactions that specify the bpA before branching. **b**, Positioning of the bpA in the yeast spliceosome C complex. **c**, Interaction network surrounding the bpA postligation. **d**, Aligned D6 helices showing conformational movement of the bpA during stages of splicing. **e**, Conformational rearrangement of D6 from branching to exon ligation. **f**–**h**, Secondary structure schematic with annotated tertiary contacts of the RNP in the pre-1F (**f**), the pre-2F (**g**) and the post-2F state (**h**). Yellow cartoon batons indicate the position of the maturase DBD helices.

the opposite side of the branch helix and points toward the 3′-end of D6 (Fig. 3d and Supplementary Video 2). This dramatic conformational rearrangement of the bpA relaxes the backbone distortion, potentially releasing free energy that compensates for the energetic cost of disengaging interactions that originally anchored the bpA[5]. Upon exon ligation, the 3′-end of the intron (C635 and G636) moves further towards the branch site and makes direct contact with the Hoogsteen face of the bpA (Fig. 3c). These extra molecular interactions, formed after exon ligation, limit the conformational flexibility of the bpA and mark the termination of the splicing pathway.

## Branch helix conformational dynamics

In addition to the local dynamics of the bpA, comparison of the intron RNP structures reveals a set of tertiary contact rearrangements needed to coordinate the two sequential steps of splicing (Fig. 3e–h and Supplementary Video 3). In the pre-1F state, the intron recognizes the 5′-exon through the EBS1–IBS1 interaction and the branch helix adopts the D1c- and maturase-docked conformation (Fig. 2a), which we refer to as the 'up' position hereafter (Fig. 3e). In this arrangement, an array of long-range interactions form between J4/5 (A559 and A560) and J5/6 (U591, G592, U593) which participate in a coordinated series of interactions (Extended Data Fig. 6c). This network begins with a canonical base pair (A560-U591) and continues with a non-canonical pairing, in which the Hoogsteen edge of A559 interacts with the sugar edge of G592 and is capped by the final nucleotide of J5/6, U593, which stacks beneath A559. Owing to the interactions that pull D6 into the up position and the constraints imposed by the J5/6 interaction network, the phosphate

backbone connecting D5 and D6 adopts a bent conformation, flipping adjacent nucleotides to opposing sides (Extended Data Fig. 6d).

After branching, D6 undergoes a substantial structural rearrangement that involves an approximately 90° swing, to the 'down' position (Fig. 3e). Through this process, the intron pulls the 5′SS and the newly formed lariat bond about 21 Å out of the active site and exchanges it for the 3′SS, thereby preparing the active site for exon ligation. During this transition, the ν–ν′ tertiary interaction and D6–maturase contacts are disrupted. In the resulting pre-2F structure, D6 docks onto D2, engaging π–π′, which latches onto the branch helix, thereby pulling D6 and the covalently linked 3′-exon into position (Fig. 3g). This allows formation of the EBS3–IBS3 base pairing that defines the 3′SS (U(+1)-A231) (Fig. 3f,g). Comparison of the catalytic D5 helix in the pre-1F and pre-2F structures reveals that it remains stationary within the D1 scaffold (root mean square deviation of 0.5 Å). Instead, movement of the D6 helix hinges on the J5/6 linker and appears as motion of the branch helix relative to a fixed RNP body. Swinging of D6 into the plane of the RNP relaxes the bent conformation of J5/6 (Extended Data Fig. 6e), enabling an exchange of substrates in the active site and driving the branching reaction forward[5]. The structural importance of J5/6 in branching is consistent with mutational studies that investigated its biochemical function in positioning of the branch helix[28,29]. Further movement of D6 is observed on completion of splicing, where there is minor motion of the D6 3′-end, which tucks inwards, allowing engagement of γ–γ′ (A327-U638) (Fig. 3h). Hence, our structures provide detailed molecular insights into the conformational rearrangements and sequential transitions that are required for branching and SS specification during group II intron splicing.

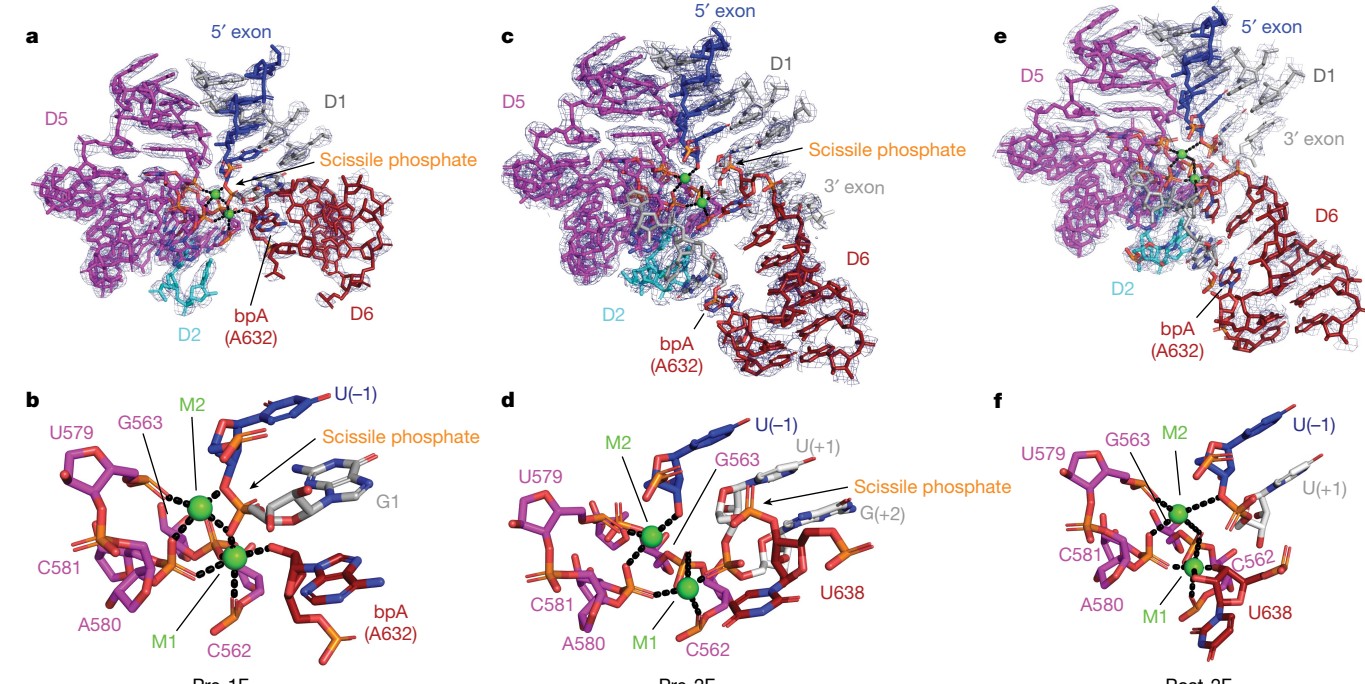

**Fig. 4 | Molecular mechanism of group II RNP branching and exon ligation. a**, Organization of catalytic elements before branching. The bpA is juxtaposed to the 5′SS and poised for lariat formation. **b**, Zoomed in view of (**a**). **c**, Active site configuration before exon ligation. The 5′SS is primed for attack to ligate the exons. **d**, Zoomed in view of (**c**). **e**, Positioning of active site elements immediately after exon ligation. **f**, Zoomed in view of (**e**). Divalent metal ions are shown as green spheres.

## Catalytic mechanism of intron splicing

Having revealed the dynamical strategies used by the group II intron RNP throughout the branching pathway, we next sought to visualize the chemical catalytic mechanisms for each step. Catalysis of the branching reaction is potentiated by a heteronuclear metal ion core organized around the catalytic triplex and the two-nucleotide bulge of D5 (Fig. 4a,b). Through precise positioning of D6 and formation of intra-D6 interactions (Fig. 2a and Fig. 3f), the first-step nucleophile (2′-OH of the bpA) is precisely positioned by catalytic metal M1 and placed in the activation distance (2.3 Å), where it is poised for the nucleophilic attack (Fig. 4b). Remarkably, the attacking 2′-OH nucleophile in the pre-1F structure occupies an identical position to that of the water nucleophile in an earlier pre-hydrolytic structure (Fig. 5a), which provides an unambiguous explanation for the competitive nature of the two splicing pathways[30]. The scissile phosphate of the 5′SS, between U(−1) and G1, adopts the same sharply kinked conformation previously observed for the hydrolytic, precatalytic state[7]. The pro-Rp oxygen of the scissile phosphate coordinates both M1 and M2 whereas the 3′-bridging oxygen is in direct contact with M2, facilitating departure of the 3′-oxyanion leaving group. This high-resolution view of the active site in the prebranching state hence provides direct visualization of the two-metal ion mechanism for group II intron branching proposed three decades ago[31]. In addition, we identified two strong, globular densities around the divalent metal core, whose positions correspond to the previously identified monovalent ions, K1 and K2 (ref. 7) (Extended Data Fig. 7a,b). Our findings therefore highlight the formation of a heteronuclear metal ion core as a general catalytic strategy fundamental to RNA splicing[6,7,32].

Upon cleavage of the 5′SS, D6 movement brings the first-step nucleophile and the now covalently linked G1 out of the active site (Fig. 3e and Fig. 4c,d). The first-step leaving group, the U(−1):3′-OH, remains tightly coordinated with catalytic metal M2 and becomes the activated second-step nucleophile. The second-step scissile phosphate between

U638 and U(+1) then becomes visible in the pre-2F state, adopting the same precleavage kinked configuration (Fig. 4c,d). These data establish that the same active site is used for both splicing steps without modifying the catalytic ion configuration nor the metal-binding platform (Fig. 4b,d and Extended Data Fig. 7). Moreover, our structure of the post-2F state with the ligated exon bound (Fig. 4e,f) shows that the metal catalytic core remains well organized, whereby the 3′-bridging oxygen of U(−1) remains associated with M2 and the 3′-OH of U638 is coordinated with M1 (Fig. 4f).

## Discussion

### Splicing at the RNP interface

As revealed by our study, group II intron and spliceosome not only share structural and chemical components (Extended Data Fig. 7c) but also a conserved dynamical strategy for sequential rearrangement between the steps of splicing. Direct parallels can be drawn between the motions of the D6 helix in the group II intron and the branch helix in the spliceosome. Comparison of their identical branching states reveals the same 90° swing of the U2-intron branch helix during the transition from the branching B* complex to the exon ligation C* complex (Fig. 5b,c). Analogous conformational dynamics are observed in the group II intron holoenzyme, as it swaps SSs without disrupting the catalytic core, when transitioning between the steps of splicing. Remarkably, the branch helix swinging motion has equivalent centres of rotation to that of the spliceosome, whereby the J5/6 linker in the group II intron acts as a hinge, much like the corresponding U2/U6 linker in the spliceosome[29]. Intriguingly, as in the spliceosome (U2/U6), there are no conformational rearrangements of the catalytic triplex (Fig. 5b,c), which remains static through the stages of branching[19]. We now have direct evidence of a conserved dynamical mechanism of SS exchange by group II introns that has direct parallels with the spliceosome, strengthening the argument that group II introns and spliceosome share the same ancestry.

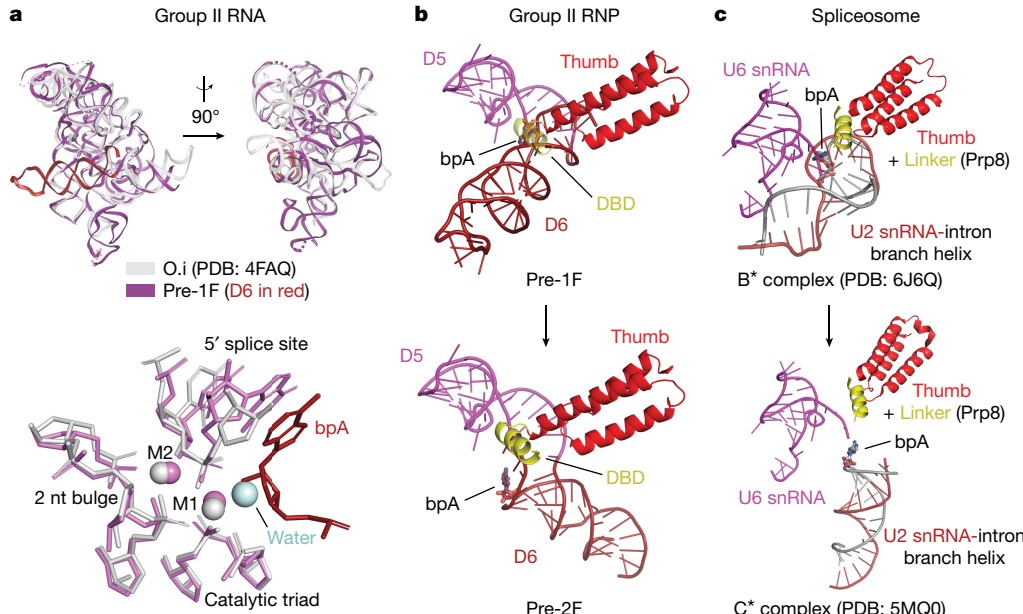

**Fig. 5 | Mechanistic comparison of group II introns and the spliceosome.**
**a**, Comparison of the overall fold of group IIC introns (top) and the aligned active sites of the *Oceanobacillus iheyensis* intron before hydrolysis (nucleophilic water in light blue) and the *E. rectale* intron before branching (bpA in dark red) (bottom). **b,c**, Conserved RNP interface and branch helix dynamics in group II RNPs (**b**) and spliceosomes (**c**) during the first- to second-step transition.

Despite the many features in common with group II introns, we identified a functional difference that provides an extra layer of regulation for the spliceosome. The first short α-helix located in the maturase DBD domain has a positively charged surface (Extended Data Fig. 8a) that is indispensable for spontaneous group II intron RNP branching (Fig. 2e,f). In contrast, whereas the equivalent helix in the linker domain of Prp8 adopts a highly similar pose (Fig. 5b,c), the contact surface is negative to neutral in charge (Extended Data Fig. 8b). This marked difference between the maturase and Prp8 has evolutionary implications. On one hand, the maturase is the lone protein cofactor necessary for proper positioning of the D6 branch helix. However, the spliceosome requires recruitment of step 1 specific factors, such as Yju2, to activate branching[19]. We can now structurally rationalize the need for Yju2 by comparing the maturase and Prp8 surfaces. The N terminus of Yju2 may compensate for the positive charges that are lost during molecular evolution from the maturase to Prp8 by forming a highly positively charged contact surface that interacts with the branch helix (Extended Data Fig. 8b). Intriguingly, the maturase side chain, Lys361, shown to be essential for intron branching in our study (Fig. 2f) has no equivalent in Prp8; whereas a highly conserved residue (Arg3 in *Saccharomyces cerevisae* and *Homo sapiens*)[33] at the N terminus of Yju2 plays a similar role in contacting G1 of the 5′SS. We therefore observe hints of a molecular evolutionary strategy that fragmented the single-protein RNP into a multiprotein splicing machine, which allows for fine tuning of RNA splicing as a regulated biological process.

### Conservation of molecular recognition

The prebranching RNP structure presented in this study reveals the 5′SS and branchpoint recognition strategy used by group II introns, thereby providing critical insights into how splicing machinery maintains precise SS and branchpoint definition during molecular evolution.

We present the pre-attack conformation of the bpA in group II introns (Fig. 4a). This high-resolution view unambiguously explains the branch site recognition strategy used by group II introns. Instead of canonical base pairing, the intron resorts to a base triple (*cis* Watson–Crick/ sugar edge interaction) formed between the bpA and a CG base pair located two nucleotides upstream (Fig. 3a). The interaction also serves

to hold the bpA inwards, toward the major groove of the D6 branch helix, thereby limiting its conformational flexibility and correctly positioning its 2′-OH relative to the catalytic metal for activation. The same molecular recognition strategy is used by the spliceosome to anchor its bpA (Fig. 3b). This striking similarity provides molecular evidence that there is minimal change to the strategy for branchpoint definition during evolution from group II introns to the spliceosome.

Also, we revealed the molecular basis of group II intron 5′SS recognition. Through base–sugar interactions originating from G1 and a base triple interaction from U2 (Extended Data Fig. 6a), the intron 5′SS interlocks with the branch helix and closely contacts the branchpoint, preparing the system for branching. The abundance of molecular interactions surrounding the 5′SS also enforces stringent nucleotide identity requirements. Given the mechanistic parallel with the spliceosome (Extended Data Fig. 6b), we can now justify why the same 5′-GU motif[24] has persisted through time, highlighting that the strategy to define the 5′SS is so robust that it has withstood the forces of molecular evolution.

### Group II RNP life cycle

By combining the cryoEM structures obtained in this study with previous mechanistic and structural work done on group II introns[7,12,17,34], we can now propose a mechanism for the group II intron splicing life cycle (Supplementary Video 4), including excision from the flanking exons and retrohoming into DNA sites (Extended Data Fig. 9 and Supplementary Video 5). After translation of the maturase from the ORF, the protein facilitates RNA folding by binding to the D4a arm and interacting with D1, which folds first and acts as a scaffold[35,36] for assembly of downstream domains. The D6 branch helix docks onto D1 through the intramolecular ν–ν′ interaction and engages the thumb/ DBD domains of the maturase to stabilize the helix in the up position. Specific molecular interactions distinguish and lock the bpA and 5′SS into place, juxtaposing the 2′-OH against the scissile phosphate for nucleophilic attack through the heteronuclear metal ion active site. During the first stage of branching, the 2′,5′-phosphodiester bond is formed, exposing the 3′-OH of the 5′-exon for ligation. To exchange substrates, the branch helix then disengages ν–ν′ and the maturase–D6 interactions, permitting the bpA to pivot around its phosphate and the

D6 helix to swing downwards (Supplementary Videos 2 and 3), where it forms the π–π′ tertiary interaction with D2. This pulls the lariat out of the active site and replaces it with the 3′SS, demarcated by the EBS3–IBS3 base pairing, thereby positioning the 3′-exon for ligation. Using the same active site, the 3′-OH is activated for nucleophilic attack, resulting in splicing of the exons. Following exon ligation, the D6 3′-tail tucks inward and the terminal nucleotide is secured by the γ−γ′ interaction. The ligated exons are then released and the liberated apoRNP retains its overall architecture, enabling it to remain primed for binding DNA substrates on the basis of shape and sequence complementarity for engagement in reverse splicing[17].

To undergo retrotransposition, we postulate that equivalent, conserved D6 and branch site motions are used[16] to achieve intron integration and substrate exchange using a persistent heteronuclear metal ion core. Given the proximity of the maturase reverse transcriptase active site to the 3′-end of the integrated intron, a logical hypothesis is that the 3′-end of the fully reverse spliced product is threaded into the maturase reverse transcriptase domain. Here, the protein, using an exogenous primer, begins target primed reverse transcription, unravelling the base pairing, disassembling the elaborate intron tertiary structure[37] and generating a complementary DNA strand to effectively copy and paste the RNA sequence into a new genomic site, thereby completing the intron life cycle. Further biochemical and structural work will be needed to evaluate this hypothesis and address the remaining mechanistic aspects of the group II RNP life cycle after branching.

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

# Methods

## Protein purification

Wild-type (WT) maturase protein (MarathonRT) was purified as previously described[17]. Briefly, the recombinant protein was cloned into a pET-SUMO vector which has an N-terminal 6×His-SUMO tag. The plasmid was transformed into Rosetta 2 (DE3) cells (MilliporeSigma), which were grown at 37 °C in LB medium supplemented with kanamycin and chloramphenicol. Cells were grown to an optical density of 1.5 in large 2 l cultures before induction with IPTG and shaking at 16 °C overnight. Cells were collected by centrifugation and resuspended in a lysis buffer (25 mM Na-HEPES, pH 7.5, 1 M NaCl, 10% glycerol and 2 mM β-mercaptoethanol (βME)) with dissolved protease inhibitor. Cells were lysed using a microfluidizer and the cell lysate was clarified to remove precipitants. The lysate was loaded onto an Ni-NTA column and was washed with lysis buffer and wash buffer (25 mM Na-HEPES, pH 7.5, 150 mM NaCl, 10% glycerol, 2 mM βME and 25 mM imidazole) before elution (25 mM Na-HEPES, pH 7.5, 150 mM NaCl, 10% glycerol, 2 mM βME and 300 mM imidazole). The SUMO tag was removed using ULP1 SUMO protease by incubating at 4 °C for 1 h. After tag cleavage, the protein was loaded onto a HiTrap SP HP cation exchange column (Cytiva) equilibrated with buffer A (25 mM K-HEPES, pH 7.5, 150 mM KCl, 10% glycerol and 1 mM dithiothreitol). The protein was eluted by running a linear gradient to buffer B (25 mM K-HEPES, pH 7.5, 2 M KCl, 10% glycerol and 1 mM dithiothreitol). The peak fractions were pooled, concentrated to 5 ml and injected onto a HiLoad 16/600 Superdex 200 pg size exclusion column (Cytiva) and eluted using a SEC buffer (50 mM $NH_4$-HEPES, 150 mM $NH_4$Cl, 5 mM dithiothreitol and 10% glycerol). Peak fractions from the S200 column were pooled, concentrated to 5 mg ml$^{-1}$, flash frozen under liquid nitrogen and stored at −80 °C. Mutant proteins (Trp310A/Gln359Ala, Lys361Ala, Lys361Ala/Thr362Ala/Asn365Ala and Lys372Ala/Arg377Ala) were also prepared in the same manner. The folding integrity of the WT maturase protein and all mutants were verified with an orthogonal activity assay involving the inherent reverse transcriptase capability[37] of these maturase proteins. All proteins were fully active for reverse transcription on long RNA templates.

## RNA transcription and purification

The DNA sequence containing the T7 promoter, 100 nucleotides (nt) of 5′ exon, the intron (with ORF deletion) and 100 nt of 3′ exon followed by the BamHI cleavage site were cloned into the pBlueScript vector (Invitrogen) to give the pLTS01 plasmid, which is used for structural studies. For splicing assays, a longer 5′ exon (150 nt) construct (which was cloned into plasmid pCZ26) was used. The plasmids were linearized using BamHI (New England Biolabs) to generate the DNA template. Mutants of the construct were prepared using a PfuUltra II Hotstart PCR Master Mix (Agilent) mixed with 20 ng of plasmid, 10 pmole of forward and reverse primers with routine site-directed mutagenesis protocol. The sequences of the resulting plasmids were verified with Sanger sequencing (Quintara Biosciences).

RNA used for cryoEM sample preparation was prepared as previously described[38]. Briefly, in vitro transcription of the intron precursor RNA was performed using inhouse-prepared T7 RNA polymerase (P266L mutant) in a transcription buffer containing 40 mM Tris-HCl pH 8.0, 10 mM NaCl, 23 mM $MgCl_2$, 2 mM spermidine, 0.01% Triton X-100, 10 mM dithiothreitol and 5 mM each rNTP. A total of 40 µg of linearized pLTS01 plasmid was added for each 1 ml of transcription and the reaction was incubated at 37 °C for 6 h before it was ethanol precipitated. The pellet was resuspended in water and mixed with an equal volume of 2× urea loading dye containing bromophenol blue and xylene cyanol, which was then loaded onto a 5% urea denaturing polyacrylamide gel to purify the intron precursor RNA. The band was visualized with ultraviolet shadowing, cut with a sterile blade, crushed with a sterile syringe and eluted in a gel elution buffer overnight (10 mM Na-MOPS pH 6.0, 300 mM NaCl and 1 mM EDTA). The eluted RNA was then ethanol precipitated, resuspended in an RNA storage buffer (6 mM Na-MES pH 6.0) to a final concentration of 50 µM and frozen at −80 °C for preparation of RNP samples.

The radiolabelled intron precursor RNA for the splicing assay was prepared using the previously described body-labelling protocol[39]. Briefly, the intron precursor RNA transcripts were body-labelled using inhouse-prepared T7 RNA polymerase and 50 µCi of [α-$^{32}$P]-UTP (PerkinElmer) in a transcription buffer containing 40 mM Tris-HCl pH 8.0, 10 mM NaCl, 15 mM $MgCl_2$, 2 mM spermidine, 0.01% Triton X-100, 10 mM dithiothreitol and 3.6 mM each rNTP (except UTP, which was at 1 mM) and 5 µg of linearized pCZ26 plasmid. Mutant intron precursor RNAs (ΔG86, ΔC601, ΔG86/ΔC601, G86A/C601U and G84A/G86A) were prepared in the same manner. The reaction was incubated at 37 °C for 1.5 h and followed by purification on a 5% urea denaturing polyacrylamide gel. The band corresponding to precursor RNA was visualized through brief exposure of a phosphor storage screen and the screen was subsequently imaged using an Amersham Typhoon RGB imager (Cytiva). The band was cut and eluted overnight in the gel elution buffer. The radiolabelled RNA was then ethanol precipitated, resuspended in the RNA storage buffer to a final concentration of 100 nM and stored in −20 °C for splicing assays.

## In vitro forward splicing assay

Radiolabelled intron precursor RNA (and mutants) were mixed with purified maturase protein (and mutants) under near-physiological condition (50 mM K-HEPES pH 7.5, 150 mM KCl and 5 mM $MgCl_2$) to perform the in vitro forward splicing assay. To do so, radiolabelled intron RNA was first mixed with buffer and water and heated to 95 °C for 1 min, after which it was returned to 37 °C for 5 min. Potassium chloride and maturase protein stocks were added and the sample was incubated at 37 °C for another 5 min. Magnesium chloride stock was subsequently added to the mixture to initiate the splicing reaction. The final concentration of radiolabelled intron precursor RNA was 5 nM and maturase protein was 20 nM. After incubation at 37 °C for 1 h, 2 µl of the reaction mixture was taken out and quenched by mixing with an equal volume of 2× formamide loading dye (72% (v/v) formamide, 10% sucrose, 0.2% bromophenol blue dye, 0.2% xylene cyanol dye and 50 mM EDTA) precooled on ice. Samples were analysed on a 5% urea denaturing polyacrylamide gel. The gel was dried and used to expose the phosphor storage screen overnight. The screen was then imaged on an Amersham Typhoon RGB imager (Cytiva). The bands were quantified using ImageQuant TL 8.2 (Cytiva).

## Group II intron RNP sample preparation

To obtain the *E. rectale* intron−maturase RNP complex stalled in the pre-1F and pre-2F states, 0.9 ml of reaction with 5 µM purified intron precursor RNA, 10 µM purified maturase protein was conducted in a buffer containing 50 mM $NH_4$-HEPES pH 7.5, 150 mM $NH_4$Cl, 10 mM $CaCl_2$ and 5 mM dithiothreitol. To do so, purified intron precursor RNA stock was mixed with buffer and water and heated to 95 °C for 3 min. It was then incubated at 37 °C for 5 min. The ammonium chloride and calcium chloride stocks were then added to refold the RNA at 37 °C for 10 min. After that, the dithiothreitol and maturase stocks were added to the reaction mixture and the reaction was incubated at 37 °C for 1 h with shaking at 300 rpm on a Thermomixer, after which it was centrifuged at 10,000g for 2 min to pellet the precipitates. The supernatant was subsequently loaded onto a Superdex 200 Increase 10/300 GL column (Cytiva) pre-equilibrated with the buffer containing 50 mM $NH_4$-HEPES pH 7.5, 150 mM $NH_4$Cl, 10 mM $CaCl_2$ and 5 mM dithiothreitol. The elution peak was pooled together and concentrated to about 4 mg ml$^{-1}$ using an Amicon concentrator (10 kDa MWCO) (MilliporeSigma). The concentrated sample was used to prepare the cryo grids (see below).

To obtain the *E. rectale* intron−maturase RNP complex in the post-2F state, the lariat intron apoRNP was first purified as previously described[17]

and the peak fraction was collected. A 1.2× molar excess of a synthetic RNA oligonucleotide with the sequence 5′-AUUUCUUUUGAAU-3′ (Integrated DNA Technologies) was then added to a final concentration of 600 nM, resulting in a final concentration of 500 nM for the RNP. The sample was incubated on ice for 10 min to allow formation of the ternary RNP complex, which was used to prepare cryo grids (see below).

All samples for cryoEM, precursor RNA and reaction ladders generated from self-splicing and maturase-mediate splicing were loaded onto a 5% urea denaturing polyacrylamide gel run in a Mini-PROTEAN tetravertical electrophoresis cell (Bio-rad) at 180 W for 50 min before staining with GelRed (Biotium) and imaging with the Cy3 channel on an Amersham Typhoon RGB imager (Cytiva).

## Grid preparation and data collection

For cryoEM analysis of the *E. rectale* intron–maturase RNP complex stalled in the pre-1F and pre-2F states, 7 μl of purified sample was loaded onto the Chameleon system (SPT Labtech). A total of 40 nl of the sample solution was dispensed to the glow-discharged Quantifoil Active Cu 300-mesh R1.2/2 grids with Cu nanowires (SPT Labtech) and the grids were plunged into liquid ethane and frozen in liquid nitrogen about 400 ms after. The grids were screened on a Talos Glacios microscope (ThermoFisher) operating at 200 keV. Grids with sufficient collectable squares and minimal crystalline ice contamination were selected for data collection on a Titan Krios microscope (ThermoFisher) operating at 300 keV with a K3 summit direct electron detector (Gatan) and the data were obtained in the counting mode. SerialEM v.3.9 was used for data collection. Two datasets of 4,612 micrographs (at 30° tilt angle) and 7,777 micrographs (at 0° tilt angle) were collected for the pre-1F/pre-2F sample. A nominal magnification of 81,000× and a defocus range of −0.8 to −2.5 μm was used, giving an effective pixel size of 0.844 Å at the specimen level. Each micrograph was dose-fractionated to 40 frames with a total exposure time of 3.482 s and a frame exposure time of 0.0865 s, resulting in a total dose of 60 e⁻/Å². 

For cryoEM analysis of the *E. rectale* intron–maturase RNP complex in the post-2F state, two separate approaches were taken. The first strategy used the Chameleon for grid preparation. Here, 8 μl of purified sample was loaded onto the Chameleon system (SPT Labtech). A total of 40 nl of the sample solution was dispensed to the glow-discharged Quantifoil Active Cu 300-mesh R1.2/2 grids with Cu nanowires (SPT Labtech) and the grids were plunged into liquid ethane and frozen in liquid nitrogen about 300 ms after. In parallel, grids were prepared using the Vitrobot. Here, 4 μl of the purified RNP sample was loaded onto plasma cleaned QuantiFoil Cu R1.2/1.3 300-mesh grids prepped inhouse with an extra layer of carbon. The grids were blotted and plunged into liquid ethane and frozen in liquid nitrogen using a condition of 100% humidity at 22 °C. SerialEM v.4.0 was used for data and micrographs were recorded on a Titan Krios microscope (ThermoFisher) operating at 300 kV equipped with a K3 Summit direct electron detector (Gatan) operating in counting mode. Two datasets of 13,740 micrographs (Chameleon) and 3,852 micrographs (Vitrobot) were collected for the post-2F sample. For the Chameleon dataset, a nominal magnification of ×105,000 and a defocus range of −1.0 to −2.5 μm was used, giving an effective pixel size of 0.832 Å at the specimen level. Each micrograph was dose-fractionated to 50 frames with a total exposure time of 1.5 s and a frame exposure time of 0.03 s, resulting in a total dose of 50.5 e⁻/Å². For the Vitrobot dataset, a nominal magnification of ×105,000 and a defocus range of −1.0 to −2.0 μm was used, giving an effective pixel size of 0.832 Å at the specimen level. Each micrograph was dose-fractionated to 48 frames with a total exposure time of 1.92 s and a frame exposure time of 0.04 s, resulting in a total dose of 50 e⁻/Å².

## CryoEM data processing

Recorded video frames were processed using cryoSPARC v.3.4 (refs. 40,41). Motion correction and contrast transfer function (CTF) estimations were performed using default parameters in cryoSPARC.

Exposures were curated and micrographs with obvious ice contamination, large motions or damaged areas were removed.

For the pre-1F and pre-2F samples, particle picking was done with the automated blob picker and filtered using consecutive rounds of two-dimensional (2D) classification. This subset of particles was used for Topaz (Topaz 0.2.4) training and picking on the combined 11,783 micrographs. Topaz picking yielded 1,289,915 particles which were subject to three rounds of 2D classification, leaving 847,534 particles which were extracted with a box size of 384 × 384 pixels. A total of 100,000 particles were selected for initial model creation, generating three reconstructions, of which one was selected as the reference for further classification. In the first round of 3D classification, eight classes were separated out, from which two groups of 422,728 particles and 234,625 particles corresponding to the pre-1F and pre-2F states were identified. Each group was subjected to another round of 3D classification, leaving a final subset of particles of 281,619 particles (pre-1F) and 234,625 particles (pre-2F, all particles were selected). Each group of particles was separately refined, yielding reconstructions of 3.0 and 3.1 Å, respectively. For each reconstruction, separate masked local refinements and local and global CTF refinement were conducted on the left and right halves to improve resolution of each. Four different maps were obtained for the left and right halves of the pre-1F and pre-2F reconstructions which have resolutions of 2.9, 3.1, 3.0 and 3.3 Å, respectively, as evaluated by the GSFSC with a cutoff of 0.143 (Extended Data Figs. 2 and 3).

A similar strategy was used to obtain the 3D reconstruction of the post-2F state. Briefly, particles were picked with an automated blob picker from the first dataset of 13,740 micrographs (Chameleon grid) and the results were filtered through several rounds of 2D classification and used as an input for Topaz training. This yielded 930,320 particles which were further cleaned through iterative rounds of 2D classification leaving 695,515 particles. Three initial models were generated with a subset of 100,000 particles out of which a single refinement with the best overall density was chosen. The 3D classification was used to separate the particles into ten classes from which a single class with 217,454 particles was selected. These particles were used to train a Topaz model and after iterative rounds of 2D classification cleaning, 361,978 particles remained. These particles were combined with 342,031 particles that were Topaz picked and filtered from a separate dataset of 3,852 micrographs (Vitrobot grid). The combined 722,394 particles were 2D classified and manually separated into groups corresponding to the dominant and less-represented views. The 70,000 randomized particles were taken from the dominant view group and aggregated with all particles from the 'other' views. Particles were extracted with a box size of 384 × 384 pixels. After an initial 3D refinement, 3D classification into ten classes yielded two classes with good overall density, which were used to generate an overall reconstruction of 3.0 Å. Similar to the other branching structures, masked local refinement was done on the separate halves along with local and global CTF refinement. This resulted in reconstructions of 2.8 and 3.2 Å for the left and right halves, respectively, as evaluated by the GSFSC with a cutoff of 0.143 (Extended Data Fig. 4).

## Model building and refinement

Model building was initiated by docking a previous group II RNP structure (PDB: 7UIN) into the generated reconstructions using UCSF Chimera[42–44] (Chimera v.1.15 and ChimeraX v.1.2.5). NAMDINATOR[45] (https://namdinator.au.dk/) was used for flexible fitting of the docked models to obtain better starting models. The models were then manually rebuilt in COOT (COOT v.0.9.6) to accommodate for the changes in branch helix position, the extra ligated exons and the metal ion core. Density for the distal portion of D6 was weak in the reconstructions but this region was modelled as a helix nonetheless on the basis of data that demonstrated that this domain forms a canonical helix. The three-way

junction of the D4 arm and portions of the α–α′ interaction were not modelled as the density is difficult to interpret. The final pre-1F, pre-2F and post-2F models were improved by iterative rounds of real-space refinement against the sharpened cryoEM map in PHENIX (Phenix v.1.20.1-4487) using secondary structure restraints for RNA, protein and DNA, as well Ramachandran and rotamer restraints for protein chains and subsequent rebuilding in COOT[46–48]. Model building and validation statistics are listed in Extended Data Table 1. Directional resolution anisotropy analyses were performed using the 3DFSC[23] web server (https://3dfsc.salk.edu/).

### Protein conservation analysis
Group IIC maturase protein sequences were obtained from the Bacterial Group II Intron Database[49] (http://webapps2.ucalgary.ca/~groupii/). Roughly 90 sequences were aligned with ClustalOmega. Alignments were analysed and visualized using JalView.

### RNA conservation analysis
Sequences corresponding to group IIC D1c and D6 sequences were obtained from the Bacterial Group II Intron Database[49] (http://webapps2.ucalgary.ca/~groupii/). Sequences were aligned with ClustalOmega and alignments were visualized and analysed with JalView.

### Figure preparation
Figures and illustrations were prepared using PyMOL (PyMOL v.2.6.0), GraphPad Prism v.9.2, RNA2Drawer[50] (https://rna2drawer.app/) and Adobe Illustrator.

### Reporting summary
Further information on research design is available in the Nature Portfolio Reporting Summary linked to this article.

## Data availability
All data are available in the main text and the Supplementary materials. CryoEM maps generated in this study are deposited in the Electron Microscopy Data Bank with codes EMD-40986 (pre-1F), EMD-40985 (pre-2F) and EMD-40987 (post-2F). Structural models are available in the Protein Data Bank with PDB accession codes 8T2S (pre-1F), 8T2R (pre-2F) and 8T2T (post-2F). Spliceosome and group II intron structural models used in this study (as an initial model for building or for comparison) are publicly available with the following PDB accession codes: 6J6Q (yeast spliceosome B* complex), 7B9V (yeast spliceosome C complex), 5MQ0 (yeast spliceosome C* complex), 7UIN (*E. rectale* group II RNP) and 4FAQ (*O. iheyensis* group II intron RNA).

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

**Acknowledgements** A.M.P. is an Investigator and L.X. is a Research Associate of the Howard Hughes Medical Institute (HHMI). We thank J. Peng for computational assistance and S. Devarkar for help with model building. We thank M. Llaguno (Yale) for assistance in Chameleon sample preparation. We thank S. Yang, X. Zhao and Z. Yu (HHMI Janelia) and S. Wu, J. Lin and K. Zhou (Yale) for assistance in cryoEM data collection. We thank W. Wang, S. Patel and O. Fedorova for insights and advice throughout this project. We thank M. Riggi and J. Iwasa (University of Utah) for their help with animating the group II intron splicing pathway. This work was supported by HHMI and the Gruber Foundation (Gruber Science Fellowship to K.C.). CryoEM data were collected with microscopes at the Yale CryoEM Resource Core that is funded in part by the NIH (S10OD023603).

**Author contributions** L.X., T.L., K.C. and A.M.P. conceived the project and designed the experiments. T.L., L.X. and K.C. performed biochemical purification of the sample and prepared cryoEM samples. L.X. and K.C. performed cryoEM data analysis and model building. L.X., T.L. and K.C. performed biochemical assays. L.X., K.C. and T.L. contributed to the structure analysis. L.X., T.L. and K.C. wrote the manuscript. L.X. and A.M.P. coordinated and supervised the project.

**Competing interests** A.M.P. is a cofounder of IntronX and RNAConnect. A.M.P. has patents on MarathonRT and its use as a biotechnological enzyme. The remaining authors declare no competing interests.

**Additional information**
**Correspondence and requests for materials** should be addressed to Ling Xu or Anna Marie Pyle.

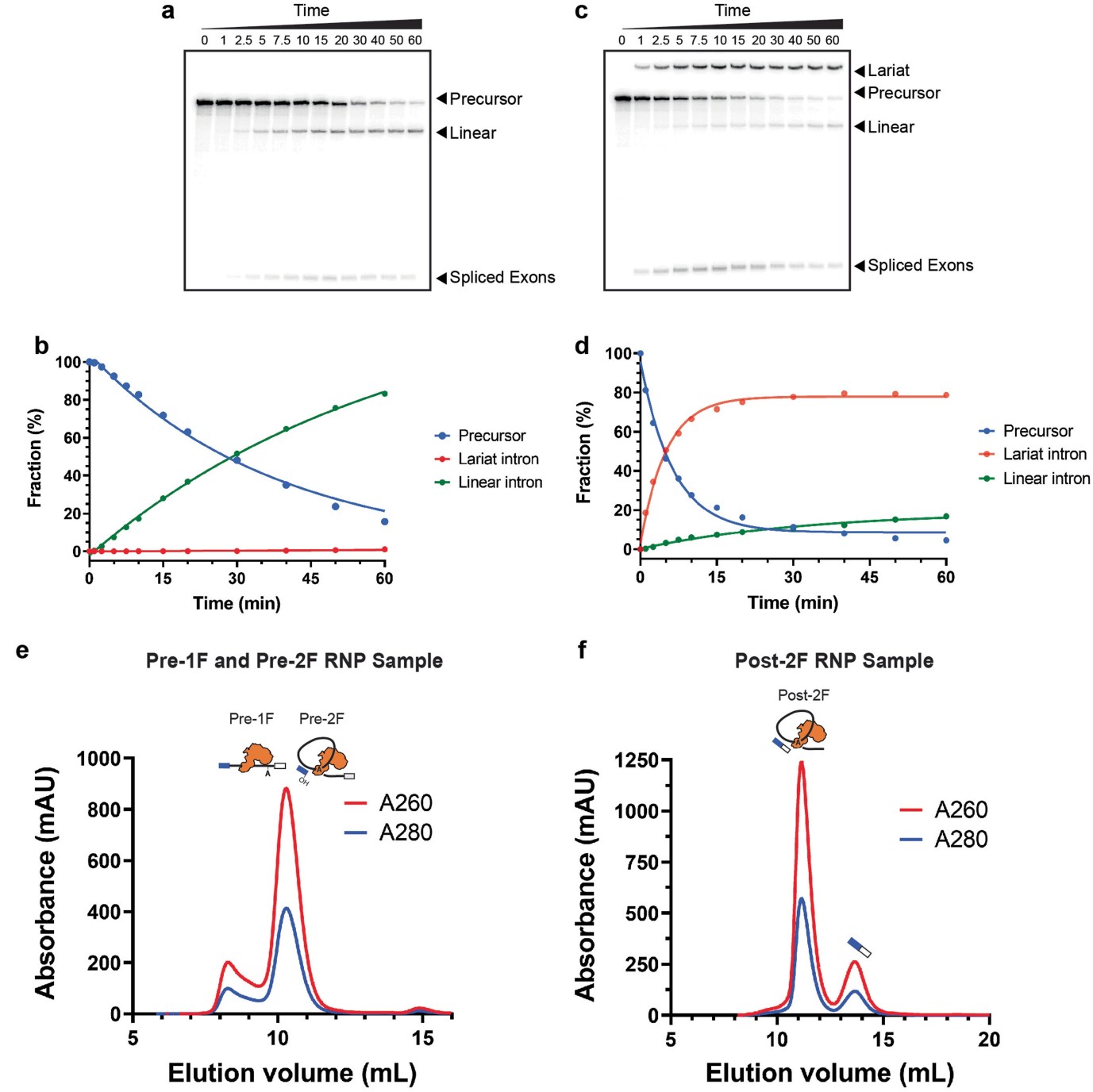

**Extended Data Fig. 1 | Biochemical investigation of group II intron branching.** The reactions are conducted under a condition that strongly favours intron branching (50 mM NH$_4$-HEPES pH 7.5, 500 mM NH$_4$Cl and 30 mM MgCl$_2$). **a.** A denaturing 5% PAGE gel showing the time course of *E.r.* intron self-splicing. The intron precursor RNA is radiolabeled with $^{32}$P uridine and the gel is imaged by autoradiography. Hydrolysis is the predominant splicing pathway, giving only linear intron. **b.** A quantitative plot of the *E.r.* intron self-splicing time course. Individual in vitro splicing time course with one independent isolated intron RNA and maturase sample is shown. **c.** A denaturing 5% PAGE gel showing the time course of *E.r.* intron splicing in the presence of the maturase protein (4-molar excess to the precursor RNA). With the facilitation of its cognate maturase protein, the major splicing pathway of *E.r.* intron is switched to branching, yielding dominantly lariat intron. Individual in vitro splicing time course with one independent isolated intron RNA and maturase sample is shown. **d.** A quantitative plot of the maturase-mediated *E.r.* intron splicing time course. The apparent precursor depletion rate increases from 0.03 min$^{-1}$ in the case of self-splicing to 0.16 min$^{-1}$ with maturase assistance. Gel filtration chromatogram of (e) the pre-1F and pre-2F and (f) the post-2F RNP sample preparation. **e.** The two peaks correspond to aggregates, co-eluted pre-1F and pre-2F RNP sample respectively. **f.** The two peaks arise from the post-2F RNP sample and the spliced exons respectively. A$_{260}$ and A$_{280}$ traces are shown in red and blue respectively.

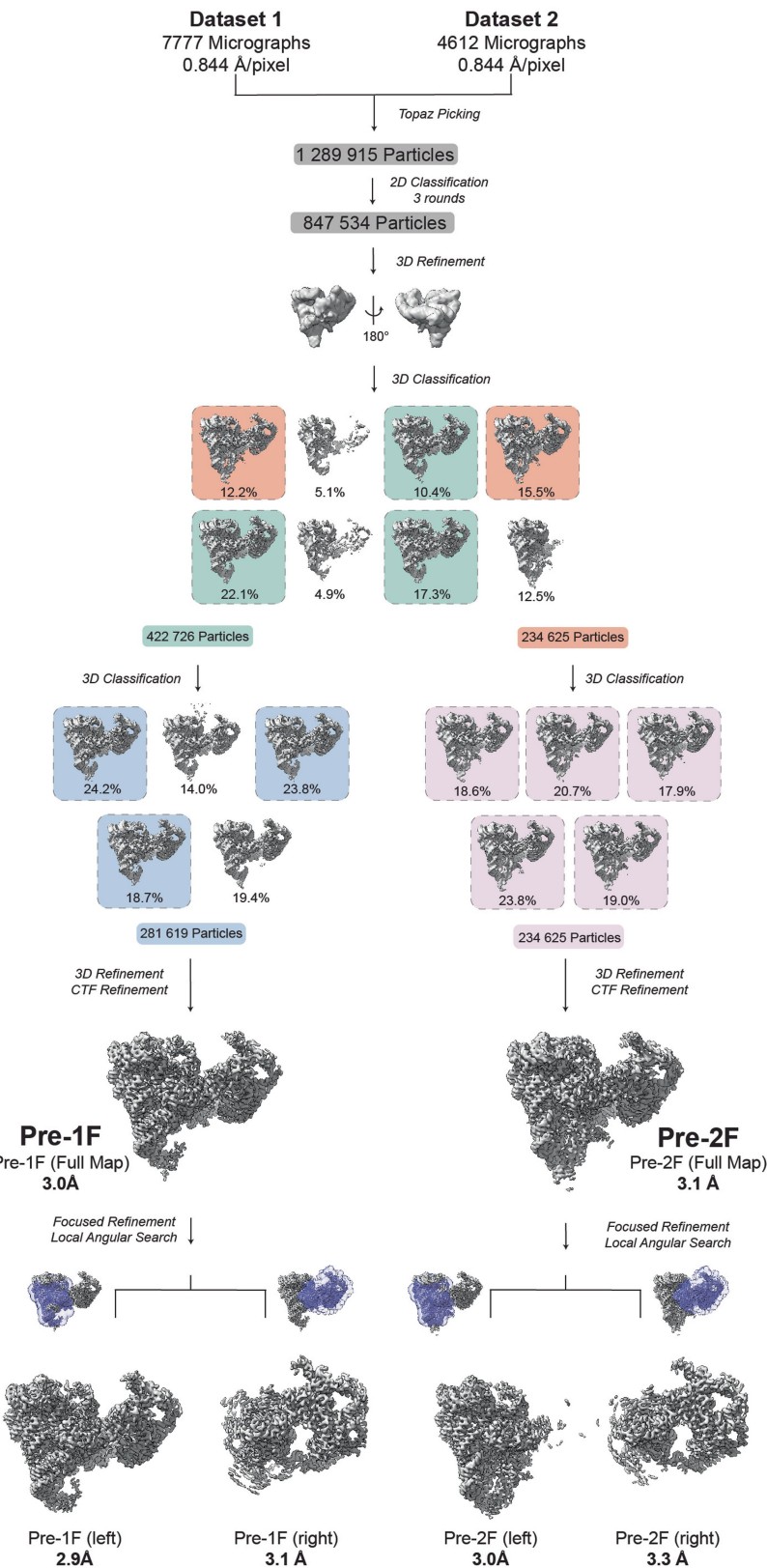

**Extended Data Fig. 2 | CryoEM Workflow for the Pre-1F and Pre-2F RNPs.** CryoEM data processing workflow of the pre-1F and pre-2F RNP samples (details in methods).

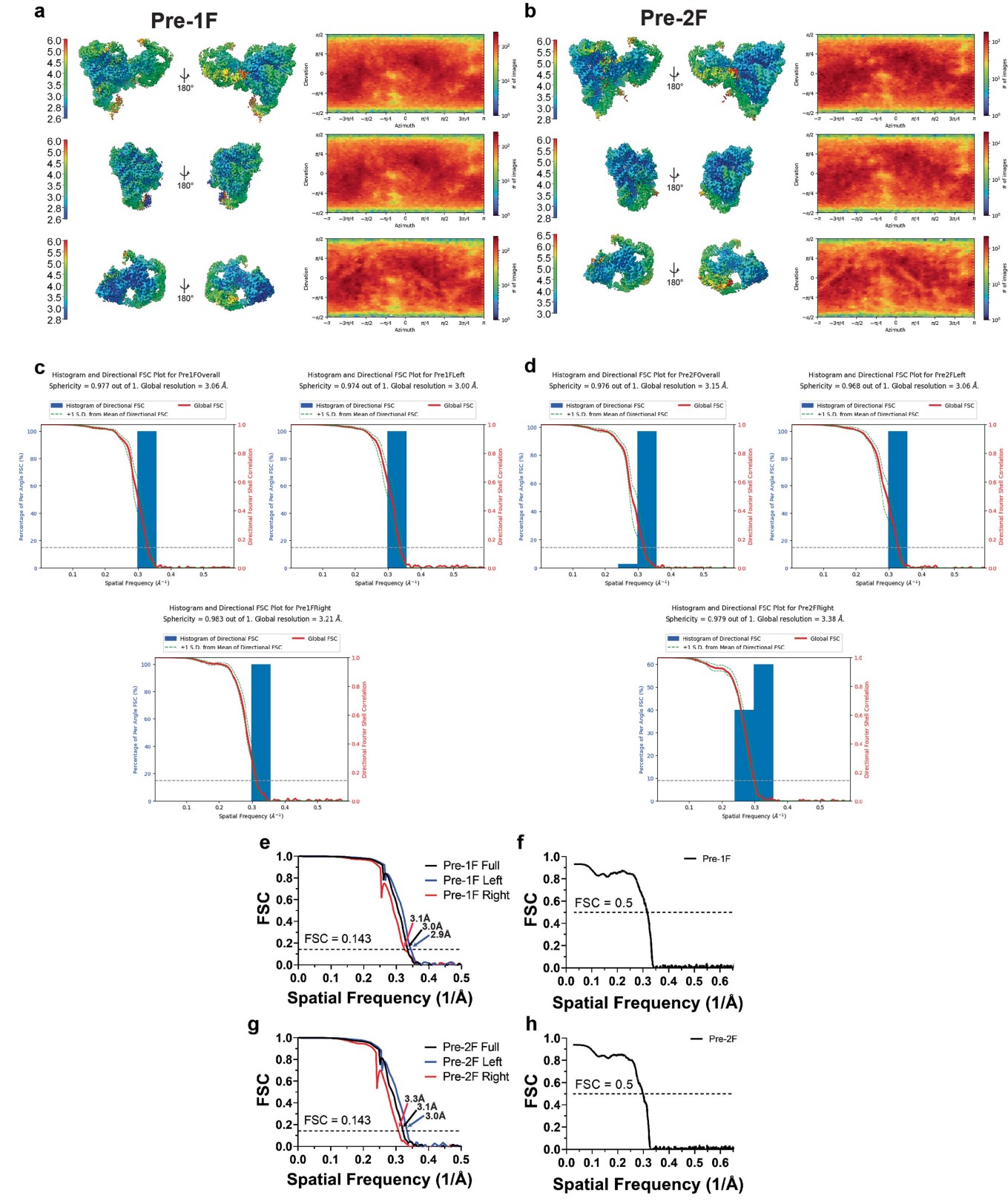

**Extended Data Fig. 3 | CryoEM Validation for the Pre-1F and Pre-2F RNPs.**
**a**. Local resolution and particle distribution of the pre-1F overall and the locally refined pre-1F (left) and pre-1F (right) maps. **b**. Local resolution and particle distribution of the pre-2F overall and the locally refined pre-2F (left) and pre-2F (right) maps. **c**. 3DFSC analysis of the pre-1F overall, the focused refined pre-1F (left) and the focused refined pre-1F (right) reconstructions. **d**. 3DFSC analysis of the pre-2F overall, the focused refined pre-2F (left) and the focused refined pre-2F (right) reconstructions. **e**. FSC curves with the gold standard threshold of 0.143 for the three pre-1F RNP maps. **f**. Map to model FSC curve for the pre-1F model refined against the pre-1F overall map. **g**. FSC curves with the gold standard threshold of 0.143 for the three pre-2F RNP maps. **h**. Map to model FSC curve for the pre-2F model refined against the pre-2F overall map.

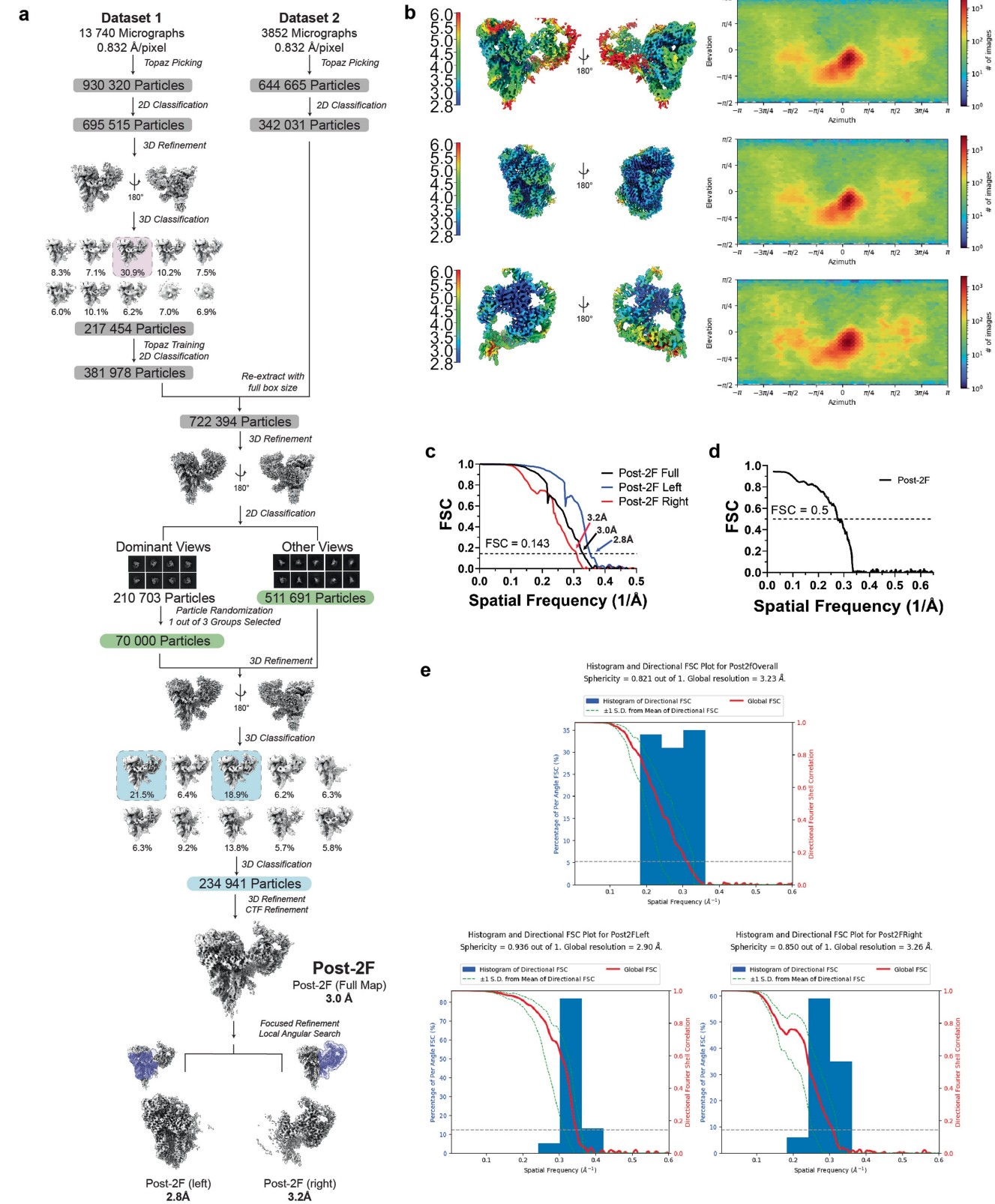

**Extended Data Fig. 4 | CryoEM Workflow for the Post-2F RNP. a**. CryoEM data processing workflow for the post-2F RNP (details in methods). **b**. Local resolution and particle distribution of the post-2F overall and the locally refined post-2F (left) and post-2F (right) maps. **c**. FSC curves with the gold standard threshold of 0.143 for the three post-2F RNP maps. **d**. Map to model FSC curve for the post-2F model refined against the post-2F overall map. **e**. 3DFSC analysis of the post-2F overall, the focused refined post-2F (left) and the focused refined post-2F (right) reconstructions.

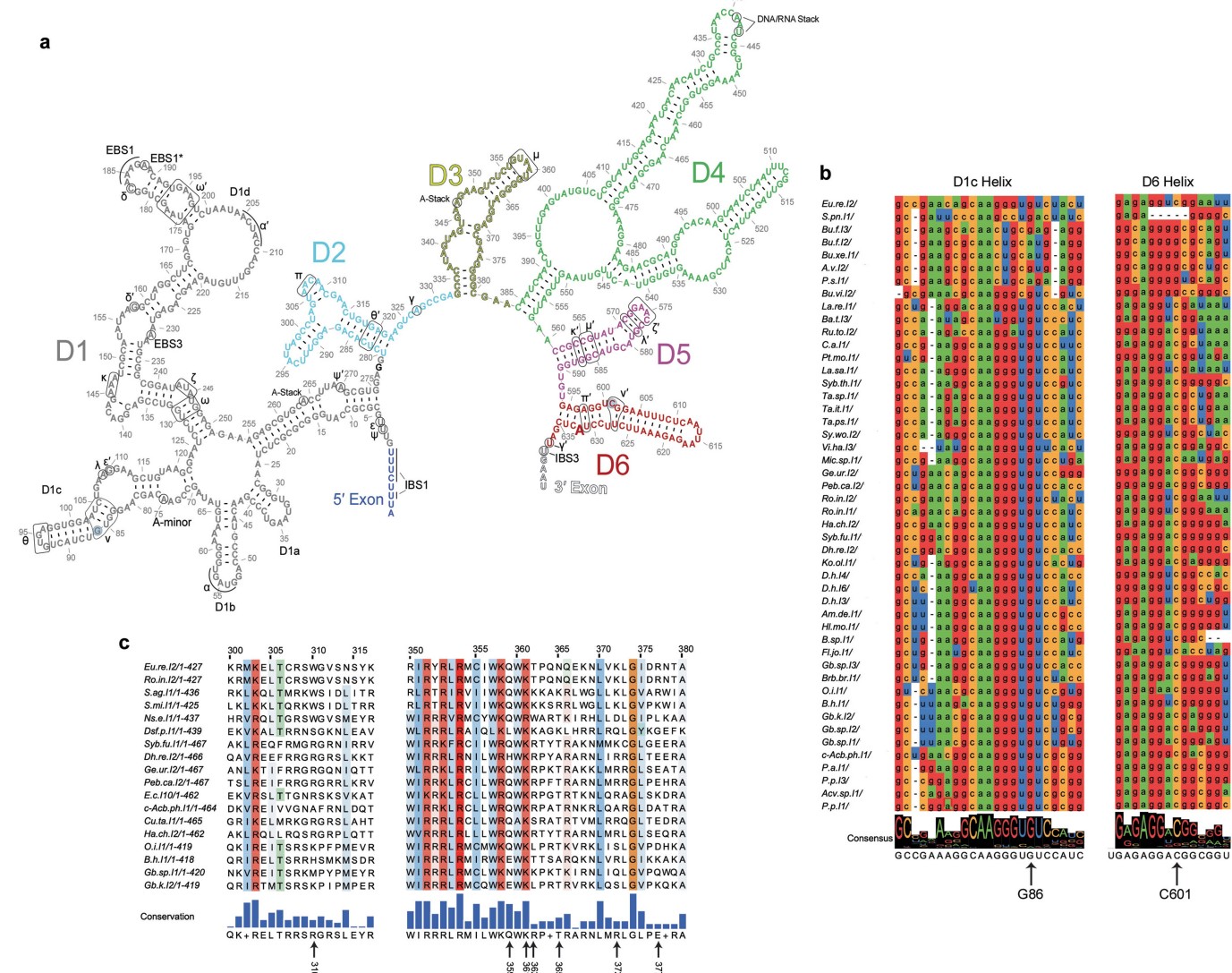

**Extended Data Fig. 5 | Secondary Structure of the *E.r.* Group IIC Intron and Conservation of Protein and RNA Sequences. a.** Secondary structure diagram of the *E.r.* IIC intron. RNA domains are labelled D1-D6 (four subdomains of D1 are labelled as D1a, D1b, D1c and D1d respectively). Tertiary interactions are labelled with Greek letters. Exon binding sequences and intron binding sequences are labelled as EBS and IBS respectively. **b**. Sequence alignment of the D1c and D6 regions of bacterial group IIC introns. **c**. Sequence alignment of bacterial group IIC maturases. The regions involved in interactions with D6 within the thumb and DBD are shown under the arrows.

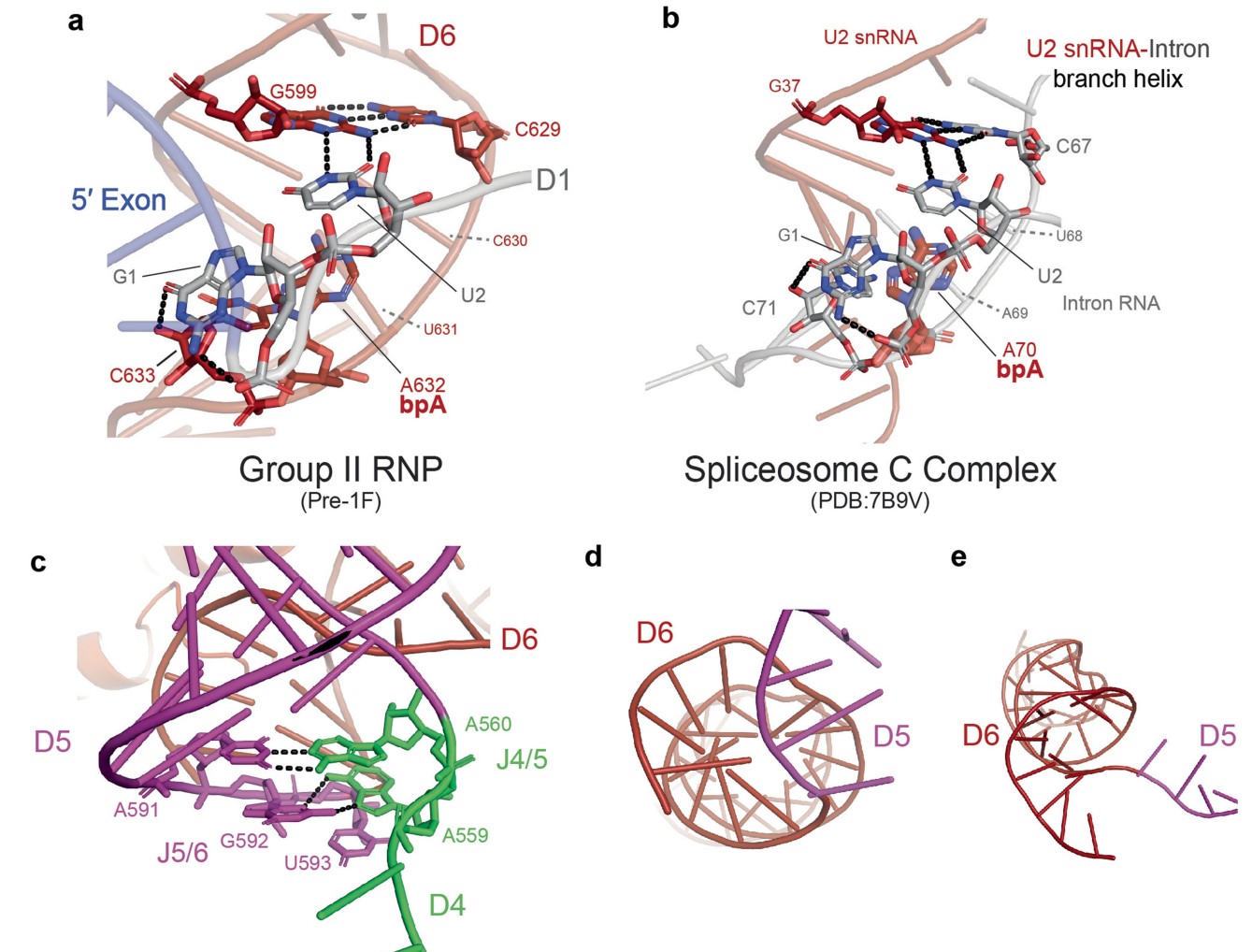

**Extended Data Fig. 6 | Mechanistic insights into 5′ Splice Site Recognition and Branch Helix Hinge Motions.** Comparison of the interactions surrounding the conserved 5′ splice site GU nucleotides for **a**. the group II RNP in the pre-1F state and **b**. the yeast spliceosome C complex (PDB:7B9V). The position of bpA adjacent nucleotides (C630 and U631 in **a**. and U68 and A69 in **b**.) are indicated with dashed lines and drawn as cartoon batons. **c**. Interaction network between J5/6 and J4/5. J5/6 RNA phosphate backbone conformations in **d**. the pre-1F and e. pre-2F states.

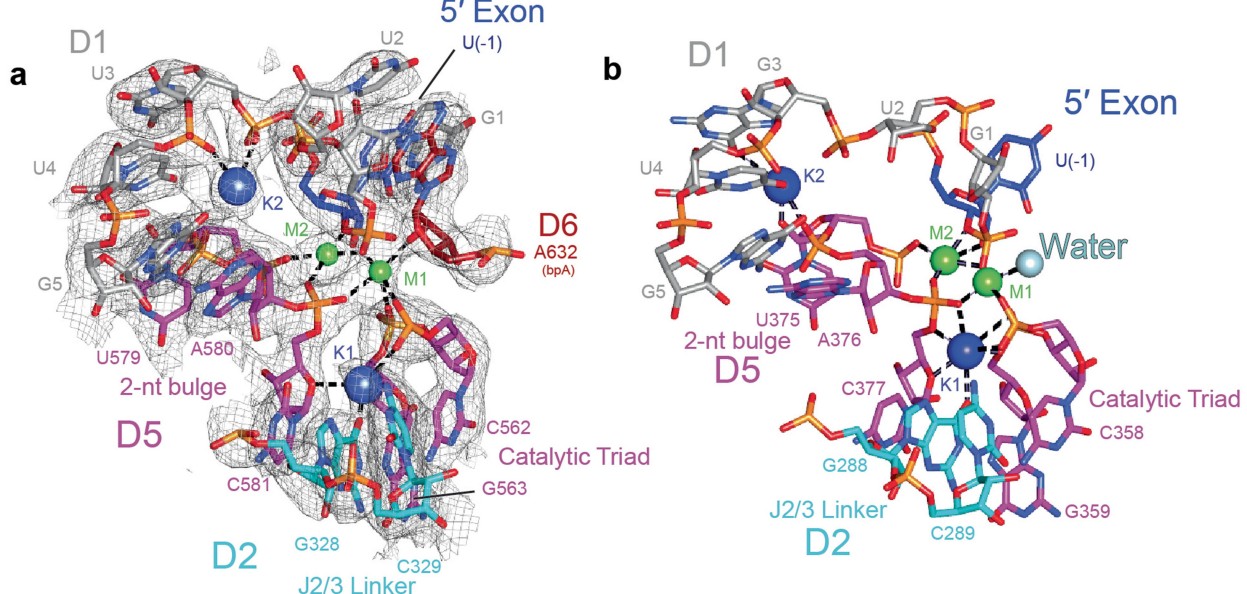

**a**
Pre-1F (*E.r.* Intron) RNP

**b**
*O.i.* Intron (PDB: 4FAQ)

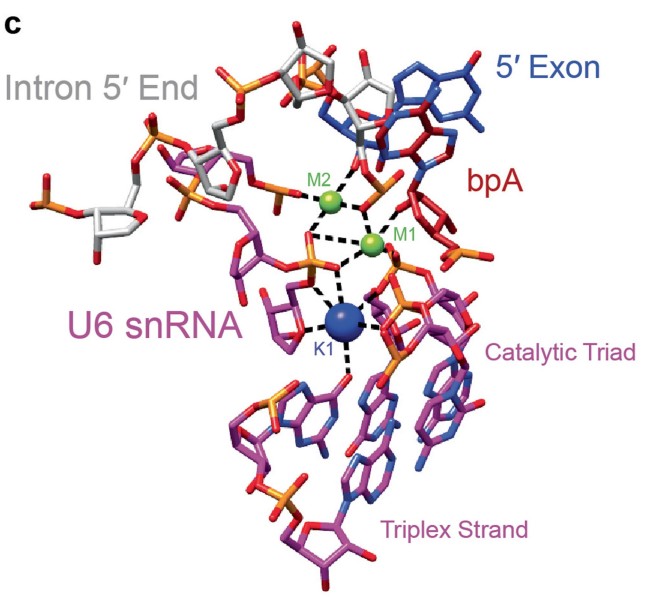

**c**
Spliceosome C Complex (PDB:7B9V)

**Extended Data Fig. 7 | Comparison of the Heteronuclear Metal Ion Core.**
**a**. The catalytic elements including the 5′ splice site, branchpoint A, J2/3 linker, catalytic triad, two-nt bulge and the metal ions (M1, M2, K1 and K2 equivalents) for the pre-1F RNP are shown. Density from the cryoEM map of the corresponding regions is shown to indicate model fit. Catalytic core of the **b**. *O.i.* intron (PDB: 4FAQ) and **c**. the spliceosome C complex (PDB:7B9V) and the key mechanistic elements as in (a) are displayed. The attacking water nucleophile is shown as a light blue sphere in (b).

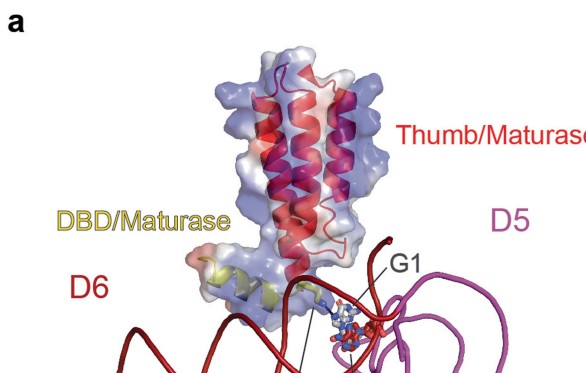

**a**

Thumb/Maturase

DBD/Maturase

D6

D5

G1

Lys361    bpA

Group II RNP
(Pre-1F)

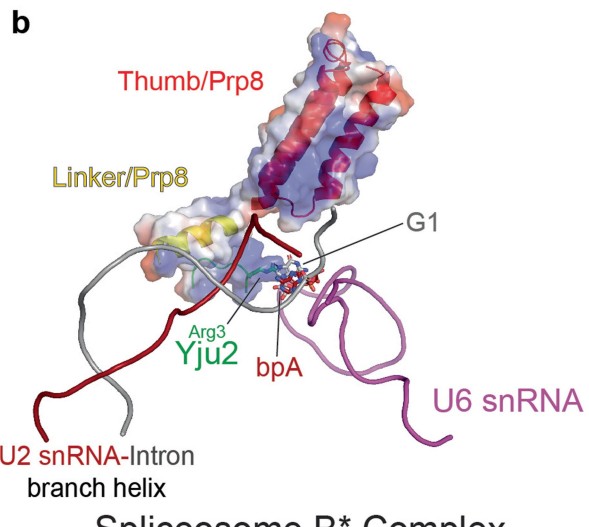

**b**

Thumb/Prp8

Linker/Prp8

G1

Arg3

Yju2    bpA

U6 snRNA

U2 snRNA-Intron
branch helix

Spliceosome B* Complex
(PDB:6J6Q)

**Extended Data Fig. 8 | Comparison of Protein-Branch Helix Interactions.**
**a.** Interactions between the DBD and thumb of the maturase protein with D6 in
the group II intron. G1 and bpA (A632) are shown in sticks. Lys361 is shown
interacting with G1. Surface charges around the protein are shown (blue-positive,
white-neutral and red-negative). **b.** Interactions between the thumb (in red)
and linker (in yellow) of Prp8 with the U2 snRNA-intron branch helix in the yeast
spliceosome B* complex (PDB: 6J6Q). Surface charges of the proteins are shown
with the same colour code as in (a). G1 and bpA (A70) are shown in sticks. Arg3
from the Yju2 protein (in green) is shown interacting with G1.

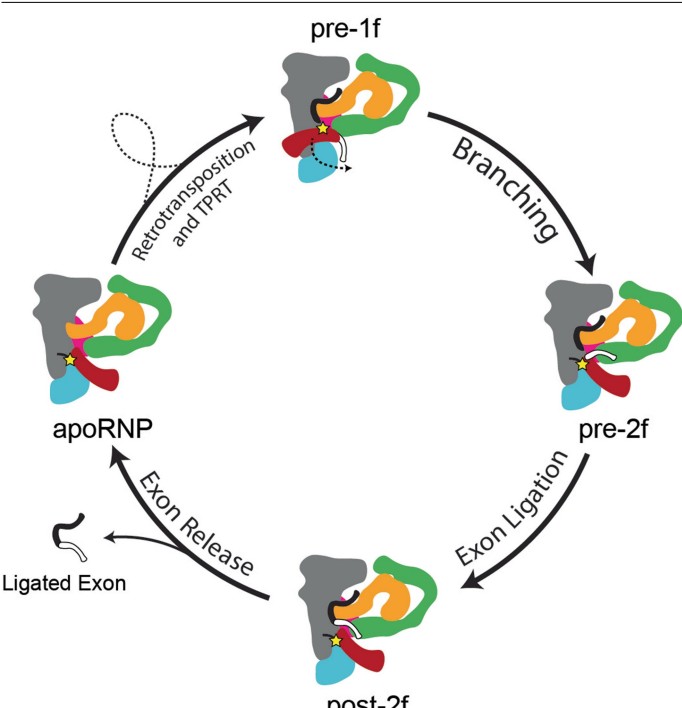

**Extended Data Fig. 9 | Model of Group II RNP Life cycle.** In the first step of splicing, the 5'-exon (in black) is recognized and juxtaposed against the branchpoint A (shown as a star) of the D6 helix. The branching reaction occurs, forming the lariat bond (shown as a dash) and the D6 helix disengages from D1c and the thumb and DBD of the maturase protein, swinging downwards and bringing the 3'-exon into the active site. The 5'-splice site is now poised to attack the scissile phosphate of the 3'-exon, splicing the exons together. After the first two steps of splicing, the ligated exon is released and the apoRNP functions as a retroelement, undergoing retrotransposition, TPRT and transcription to complete the group II intron life cycle.

**Extended Data Table 1 | CryoEM Data Collection, Refinement and Validation Statistics**

| | #1 Pre-1F (pre-branching) (EMDB-40986) (PDB 8T2S) | #2 Pre-2F (pre-ligation) (EMDB-40985) (PDB 8T2R) | #3 Post-2F (post-ligation) (EMDB-40987) (PDB 8T2T) |
|---|---|---|---|
| **Data collection and processing** | | | |
| Magnification | 81,000 | 81,000 | 81,000 |
| Voltage (kV) | 300 | 300 | 300 |
| Electron exposure (e–/Å$^2$) | 60 | 60 | 50 |
| Defocus range (μm) | -0.8 to -2.5 | -0.8 to -2.5 | -1.0 to -2.0 |
| Pixel size (Å) | 0.844 | 0.844 | 0.832 |
| Symmetry imposed | C1 | C1 | C1 |
| Initial particle images (no.) | 1,289,915 | 1,289,915 | 1,574,985 |
| Final particle images (no.) | 261,619 | 234,625 | 234,941 |
| Map resolution (Å) | 3.0 | 3.1 | 3.0 |
| FSC Threshold | 0.143 | 0.143 | 0.143 |
| | | | |
| **Refinement** | | | |
| Initial model used (PDB code) | 7UIM, 7UIN | 7UIM, 7UIN | 7UIM, 7UIN |
| Model resolution (Å) | 3.2 | 3.4 | 3.6 |
| FSC Threshold | 0.5 | 0.5 | 0.5 |
| Map sharpening $B$ factor (Å$^2$) | -107.6 | -110.6 | -74 |
| Model composition | | | |
| Non-hydrogen atoms | 15,001 | 15,066 | 14,879 |
| Protein residues | 389 | 388 | 387 |
| Ligands | 551 | 552 | 546 |
| $B$ factors (Å$^2$) | | | |
| Protein | 83.93 | 83.32 | 125.14 |
| Nucleotide | 110.23 | 110.61 | 132.45 |
| R.m.s. deviations | | | |
| Bond lengths (Å) | 0.004 | 0.006 | 0.006 |
| Bond angles (°) | 0.888 | 0.956 | 0.974 |
| Validation | | | |
| MolProbity score | 1.56 | 1.47 | 1.55 |
| Clashscroe | 3.85 | 4.24 | 4.25 |
| Ramachandran plot | | | |
| Favored (%) | 97.13 | 96.60 | 95.01 |
| Allowed (%) | 2.87 | 2.88 | 4.46 |
| Disallowed (%) | 0.00 | 0.52 | 0.52 |

Data collection refinement and validation statistics of the prebranching (pre-1F), preligation (pre-2F) and postligation (post-2F) structures.

# Reporting Summary

## Statistics

For all statistical analyses, confirm that the following items are present in the figure legend, table legend, main text, or Methods section.

| n/a | Confirmed | |
|---|---|---|
| ☒ | ☐ | The exact sample size (*n*) for each experimental group/condition, given as a discrete number and unit of measurement |
| ☐ | ☒ | A statement on whether measurements were taken from distinct samples or whether the same sample was measured repeatedly |
| ☒ | ☐ | The statistical test(s) used AND whether they are one- or two-sided<br>*Only common tests should be described solely by name; describe more complex techniques in the Methods section.* |
| ☒ | ☐ | A description of all covariates tested |
| ☒ | ☐ | A description of any assumptions or corrections, such as tests of normality and adjustment for multiple comparisons |
| ☒ | ☐ | A full description of the statistical parameters including central tendency (e.g. means) or other basic estimates (e.g. regression coefficient) AND variation (e.g. standard deviation) or associated estimates of uncertainty (e.g. confidence intervals) |
| ☒ | ☐ | For null hypothesis testing, the test statistic (e.g. *F*, *t*, *r*) with confidence intervals, effect sizes, degrees of freedom and *P* value noted<br>*Give P values as exact values whenever suitable.* |
| ☒ | ☐ | For Bayesian analysis, information on the choice of priors and Markov chain Monte Carlo settings |
| ☒ | ☐ | For hierarchical and complex designs, identification of the appropriate level for tests and full reporting of outcomes |
| ☒ | ☐ | Estimates of effect sizes (e.g. Cohen's *d*, Pearson's *r*), indicating how they were calculated |

*Our web collection on statistics for biologists contains articles on many of the points above.*

## Software and code

Policy information about availability of computer code

| | |
|---|---|
| Data collection | SerialEM v3.9 and v4.0 (cryoEM data were collected on a Thermo Fisher Titan Krios microscope equipped with a Gatan G3 Summit Detector) |
| Data analysis | All software are publicly accessible. Data processing: cryoSPARC v3.4, Topaz 0.2.4 and GraphPad Prism 9.2; Visualization: UCSF Chimera 1.15 and ChimeraX 1.2.5, PyMOL v2.6; Model building, refinement and validation: Coot 0.9.6, Phenix 1.20.1-4487, NAMDINATOR, 3DFSC; Secondary Structure: RNA2Drawer (RNACanvas), Adobe Illustrator |

For manuscripts utilizing custom algorithms or software that are central to the research but not yet described in published literature, software must be made available to editors and reviewers. We strongly encourage code deposition in a community repository (e.g. GitHub). See the Nature Portfolio guidelines for submitting code & software for further information.

## Data

Policy information about availability of data

All manuscripts must include a data availability statement. This statement should provide the following information, where applicable:
- Accession codes, unique identifiers, or web links for publicly available datasets
- A description of any restrictions on data availability
- For clinical datasets or third party data, please ensure that the statement adheres to our policy

All data are available in the main text and the supplementary materials. CryoEM maps generated in this study are deposited in the Electron Microscopy Data Bank with codes EMD-40986 (pre-1F), EMD-40985 (pre-2F), and EMD-40987 (post-2F), Structural models are available in the Protein Data Bank with PDB accession codes

8T2S (pre-1F), 8T2R (pre-2F), and 8T2T (post-2F). Spliceosome and group II intron models used for in this study (as an initial model for building or for comparison) are publicly available with the following PDB accession codes: 6J6Q (yeast spliceosome B* complex), 7B9V (yeast spliceosome C complex), 5MQ0 (yeast spliceosome C* complex), 7UIN (E.r. group II RNP), and 4FAQ (O.i. group II intron).

# Research involving human participants, their data, or biological material

Policy information about studies with human participants or human data. See also policy information about sex, gender (identity/presentation), and sexual orientation and race, ethnicity and racism.

| | |
|---|---|
| Reporting on sex and gender | N/A |
| Reporting on race, ethnicity, or other socially relevant groupings | N/A |
| Population characteristics | N/A |
| Recruitment | N/A |
| Ethics oversight | N/A |

Note that full information on the approval of the study protocol must also be provided in the manuscript.

# Field-specific reporting

Please select the one below that is the best fit for your research. If you are not sure, read the appropriate sections before making your selection.

☒ Life sciences   ☐ Behavioural & social sciences   ☐ Ecological, evolutionary & environmental sciences

For a reference copy of the document with all sections, see nature.com/documents/nr-reporting-summary-flat.pdf

# Life sciences study design

All studies must disclose on these points even when the disclosure is negative.

| | |
|---|---|
| Sample size | No statistical method was used to determine sample size. CryoEM sample sizes were determined by available microscope time and the particle density on electron microscopy grids. The sample size was sufficient to obtain structures at the reported resolution, as assessed by Fourier Shell Correlation (cutoff of 0.143). Four replicates were performed for the radioactive in vitro splicing assay with all mutant intron constructs and mutant maturase proteins. In vitro splicing time courses were done in duplicates for both conditions with and without the maturase protein. |
| Data exclusions | During cryoEM data processing, particles were excluded using standard classification approaches in cryoSPARC to remove false picks and low-resolution particle images. No data were excluded for the in vitro splicing assay. |
| Replication | All cryoEM structures were determined from independent half datasets, which were compared to determine the resolution of the respective reconstructions. It is not necessary to replicate cryoEM experiments and no replication was performed. The in vitro splicing assay with intron and maturase mutants was performed in four replicates and all attempts were successful and documented in the supplementary information. Time courses of intron splicing with and without the maturase protein were duplicated on different days with two different batches of radiolabeled RNA and independently purified maturase protein and yielded consistent banding patterns and rate constants. |
| Randomization | For cryoEM data processing, datasets are randomly split into two halves based on standard approaches in cryoSPARC. Randomization is not applicable for in vitro biochemical experiments. |
| Blinding | Blinding is not applicable to the in vitro biochemical or the cryoEM experimentation involved in this study. |

# Reporting for specific materials, systems and methods

We require information from authors about some types of materials, experimental systems and methods used in many studies. Here, indicate whether each material, system or method listed is relevant to your study. If you are not sure if a list item applies to your research, read the appropriate section before selecting a response.

## Materials & experimental systems

| n/a | Involved in the study |
|---|---|
| ☒ | Antibodies |
| ☒ | Eukaryotic cell lines |
| ☒ | Palaeontology and archaeology |
| ☒ | Animals and other organisms |
| ☒ | Clinical data |
| ☒ | Dual use research of concern |
| ☒ | Plants |

## Methods

| n/a | Involved in the study |
|---|---|
| ☒ | ChIP-seq |
| ☒ | Flow cytometry |
| ☒ | MRI-based neuroimaging |

