## [Peer Review File · Nature]

Manuscript Title: Structural insights into intron catalysis and dynamics during splicing **Reviewer Comments & Author Rebuttals**

Reviewer Reports on the Initial Version:

Referees' comments:

Referee #1:

In this manuscript the authors present three novel, high resolution cryo-EM structures of a pre-mRNA group II intron bound by the maturase protein at different stages of the splicing reaction, namely at the pre-branching, pre-ligation and post-ligation stages. The pre-branching structure reveals for the first time how domain 1 of the group II intron, together with the maturase protein, align the branch helix (D6) for step 1 of the splicing reaction in which a branched intron intermediate is formed. It elucidates the molecular interactions involving D1 nucleotides and residues of the thumb/DBD domains of the maturase protein that mediate positioning of D6. The authors carry out mutational analyses that support the functional importance of these nucleotides/amino acids, as they are shown to lead to branching defects. This structure also reveals a role for the 5'ss in docking D6 and bringing the branchpoint close to the 5'ss, which is a prerequisite for the subsequent branching reaction. Finally, the pre-branching structure elucidates the structural basis for recognition of the branchpoint adenosine. Comparisons with branchpoint recognition in the spliceosome intriguingly show that both splicing systems employ the same molecular recognition pattern, revealing a novel mechanistic parallel between the spliceosome and group II introns. The authors next compare the cryo-EM structures of pre-branching and pre-and post-ligation complexes, and thereby uncover the local conformational dynamics of the branch point adenosine during splicing. In addition, they show that D6 undergoes a major structural rearrangement after branching, swinging downward by ca 90 degrees. This repositioning removes the 5'ss and branched adenosine from the active site, and thereby allows binding instead of the 3'ss, which is a prerequisite for exon ligation. Comparisons with the published cryo-EM structures of B*, C and C* spliceosomal complexes intriguingly demonstrate an analogous movement of the U2/branch helix, which also allows docking of the 3'ss into the spliceosome's active site.

The cryo-EM structures presented in this manuscript also allow direct visualization of the two-metal ion mechanism for group II intron branching and confirm that the same active site with the same catalytic ion conformation is also used for both branching and exon ligation. They thus reveal that both the group II intron splicing machinery and the spliceosome swap the bound splice sites between the first and second catalytic steps without disrupting the catalytic core.

Finally, structural comparisons with the yeast spliceosome also reveal some intriguing differences in the roles of proteins involved in branching, and indicate that the multiple functions of the maturase protein in group II intron branching, have been fragmented and are carried out by multiple proteins (i.e., PRP8 and YJU2) in the spliceosome, which allows for a more highly regulated process.

In summary, this manuscript presents a plethora of new information about the molecular mechanisms of group II intron splicing and, in combination with previously reported group II structures, allows a complete structural and mechanistic view throughout the group II intron splicing lifecycle. Furthermore, the structural analyses presented in this manuscript provide for the first time molecular insights into the sequential structural rearrangements that are required for the branching reaction during group II splicing, as well as and for the selection of the splice sites for subsequent exon ligation. They also elucidate additional, previously unknown molecular recognition mechanisms and functionally significant structural dynamics that are evolutionarily conserved between the group II intron splicing machinery and the spliceosome, thereby strengthening the conclusion that the spliceosome has evolved from pre-mRNA group II introns.

Referee #2:

The manuscript, "Structural insights into intron catalysis and dynamics during splicing", by Xu et al. represents a significant advance in understanding the catalytic mechanism of a group II intron, particularly the new cryo-EM structure of an intron maturase complex captured just prior to the branching reaction. The structures of the same complex remodeled for exon ligation and a product complex mimic enable the authors to model the atomic interactions and rearrangements that accompany the splicing reaction. They describe structural parallels with the same reaction in pre-mRNA splicing by the spliceosome, which further reinforces support for a common evolutionary splicing ancestor. Beyond the group II intron research field, the work will be of broad interest to scientists interested in splicing mechanism, evolution of splicing, RNA structure, and RNA/protein interactions. I have no major concerns with the data and its interpretation. Minor concerns can be addressed by simple revision of some text and figures.

Specific critiques:

1. The abbreviations Pre-1F, Pre-2F and Post-2F are not intuitive. Why "F"? Avoiding the alphabet soup of the spliceosome complexes with names like, for example, "pre-branching state" or "pre-ligation complex" would be most helpful to readers.
2. The authors should take care to emphasize that current group II introns and the spliceosome are thought to share a common ancestor molecule. Although that molecule likely more closely resembles an autocatalyzing intron, statements like "living fossil" confuses the evolutionary trajectory.
3. While the ΔG_{86} and ΔC_{601} data in Fig. 2d supports the importance of both residues for branching, the argument that the G86A/C601U mutation demonstrates the importance of base pairing is overreaching. Deletion of nucleotides may have other structural effects relative to simply changing the base.
4. The large rearrangement of helix D6 following branching is striking. While Fig. 3 and the supplemental movies help to visualize the changes, it would also be nice to have a movie that highlights the interactions that must be broken to achieve the rearrangement. Can the authors further speculate what might drive this arrangement in relation to the correlating change in the spliceosome?

5. All yellow labels in figures need to be changed to black or something with enough contrast to be legible.

6. When comparing the group II intron to spliceosome structures, it would be helpful if the position of intron nucleotides relative to the branch point adenosine could also be indicated. The spliceosome intron nucleotide numbering does not make this comparison intuitive. There are also labeling errors in Ext. Data Fig. 5.5 for both U2 snRNA and the intron. Labeling the snRNA outright will help with confusion between the second U of the intron and the snRNA.

7. The resolution of Ext. Data Figs 2 and 3 needs to be improved. Currently, none of the graph labels in panels b-d are legible.

Referee #3:

This manuscript reports cryoEM structures of a group IIC intron in complex with the associated maturase protein. Group II introns are ancestral prototypes for spliceosomes, offering a fundamental model for understanding RNA splicing and the evolutionary progression of spliceosomes from retroelements. Prior endeavours to capture Group II intron complexes that elucidate the branching pathway had proven ineffective. Additionally, the role of the maturase in facilitating branching remained elusive. Both of these aspects are of paramount importance, as branching is evolutionary conserved, and the maturase stands as a vestige of the spliceosome protein PRP8.

Here, changes in the preparative biochemistry, particularly the incubation with the maturase that promotes branching, and the replacement of Mg²⁺ with Ca²⁺ enabled stalling precursor and branching intermediates for cryoEM reconstruction.

Key findings include:

The mechanism by which the maturase favors branching instead of hydrolysis, by holding D6 in an orientation that brings the branch point adenosine (bpA) to the catalytic center.

The branching mechanism involves recognizing and positioning the bpA, and using a lysine finger (K361), which binds the 5'SS to juxtapose the nucleophile with the scissile phosphate.

The substrates-exchange mechanism, by a large-scale swinging of D6 that pulls the lariat and replaces it with the 3'SS.

Novel lines of evidence for the common ancestry of spliceosomes and group II introns, with respect to the dynamics of RNA-RNA and RNA-protein contacts during branching.

Some findings are entirely unexpected, while others elucidate mechanisms proposed decades ago. The data are of high quality, and the findings hold significance for a wide readership interested in RNA metabolism and evolutionary biology. Considering these merits, I recommend the publication of this work in Nature after addressing the specific points outlined below.

Specific comments

1. The mechanistic comparison shows that the UP and DOWN positions of the D6 intron domain have equivalents in the branch duplex positions in B* and C complexes (Fig. 5b,c). The D6 domain is kept in the UP position by extensive interactions with D1 and the maturase domains (Fig. 2). Which spliceosome components interact with the branch duplex in B* and C complexes to stabilize the UP and DOWN positions, other than the Thumb-DBD of the maturase? An analysis might extend the insight into the evolution of spliceosomes, where the absence of D1 can be compensated by stage-specific proteins and PRP8 domains absent in the maturase.
2. Is it conceivable that K361 of the maturase complements the catalytic site by neutralizing charges developed during the phosphoryl transfer, upon branching? Can minor local rearrangements possibly place K361 in the appropriate distances and geometry to influence catalysis?
3. The structure-guided mutations in the maturase have a drastic effect on lariat formation (Fig. 2f, lanes 2-4). It is important to check experimentally that the mutations do not disrupt the native fold of the maturase.
4. Fig. 1d requires a more detailed description, possibly in the results section or the legend. As the maturase is present in lanes 3-5, the difference between the three samples is unclear. I understand the post-ligation RNP from lane 4 contains an oligonucleotide equivalent to the ligated exon. Are there any other differences? Besides, an additional band below the precursor is not indicated.
5. The gels from Fig. 2d,f look rather different than 1d, although both variants denote splicing gels. The linear fragment and the linear-3'exon are not indicated in 1d.
6. Fig. 2a looks overcrowded in the central area. Perhaps it will be easier to discern the various elements by removing the density (or by other means). The bpA should be indicated for the sake of orientation. I suggest indicating how the detailed views from Fig. 2b,e relate to the overview from 2a.
7. Labeling the domains in the figures that show details of interacting residues, such as Figs 2b,e, 3a-c, facilitates the analysis (e.g., D6, exon, U2 snRNA, etc).
8. A figure showing clearly the contacts between D6 and the maturase contacts relative to the bpA (A632) would be helpful. The ED Fig. 8a is useful, but insufficient.
9. Label the conformational states in Fig 3e. Fig. 3d might require optimization for conveying a message, as the indicated changes are hard to follow.
10. To better appreciate the maturase location and impact on the RNA structure, I would indicate its location (possibly as a shadow) in Fig. 3f or a similar figure from the Ext. Data.
11. A comparative view between pre-1F and the spliceosome, as in Fig. 5a bottom is required, especially since the parallels between the branching in the two splicing systems are a highlight of the paper.
12. The branch helix is not labeled in Fig. 5 and Ext. Data Fig. 8. Only the U2 snRNA is indicated, leading to the potential confusion that lone U2 is the equivalent of D6.

13. It is not clear enough the extent to which the post-2F state differs from previously published structures of group II introns. This should be indicated in the manuscript.

14. The title's reference to "intron catalysis" is ambiguous, potentially obscuring whether it relates to pre-mRNA splicing or group II introns. Given the paper's emphasis on a group II intron mechanism, a more specific title would enhance clarity. The authors may wish to double-check this aspect.

Author Rebuttals to Initial Comments:

Referee #1:

In this manuscript the authors present three novel, high resolution cryo-EM structures of a pre-mRNA group II intron bound by the maturase protein at different stages of the splicing reaction, namely at the pre-branching, pre-ligation and post-ligation stages. The pre-branching structure reveals for the first time how domain 1 of the group II intron, together with the maturase protein, align the branch helix (D6) for step 1 of the splicing reaction in which a branched intron intermediate is formed. It elucidates the molecular interactions involving D1 nucleotides and residues of the thumb/DBD domains of the maturase protein that mediate positioning of D6. The authors carry out mutational analyses that support the functional importance of these nucleotides/amino acids, as they are shown to lead to branching defects. This structure also reveals a role for the 5'ss in docking D6 and bringing the branchpoint close to the 5'ss, which is a prerequisite for the subsequent branching reaction. Finally, the pre-branching structure elucidates the structural basis for recognition of the branchpoint adenosine. Comparisons with branchpoint recognition in the spliceosome intriguingly show that both splicing systems employ the same molecular recognition pattern, revealing a novel mechanistic parallel between the spliceosome and group II introns. The authors next compare the cryo-EM structures of pre-branching and pre-and post-ligation complexes, and thereby uncover the local conformational dynamics of the branch point adenosine during splicing. In addition, they show that D6 undergoes a major structural rearrangement after branching, swinging downward by ca 90 degrees. This repositioning removes the 5'ss and branched adenosine from the active site, and thereby allows binding instead of the 3'ss, which is a prerequisite for exon ligation. Comparisons with the published cryo-EM structures of B*, C and C* spliceosomal complexes intriguingly demonstrate an analogous movement of the U2/branch helix, which also allows docking of the 3'ss into the spliceosome's active site.

The cryo-EM structures presented in this manuscript also allow direct visualization of the two-metal ion mechanism for group II intron branching and confirm that the same active site with the same catalytic ion conformation is also used for both branching and exon ligation. They thus reveal that both the group II intron splicing machinery and the spliceosome swap the bound splice sites between the first and second catalytic steps without disrupting the catalytic core.

Finally, structural comparisons with the yeast spliceosome also reveal some intriguing differences in the roles of proteins involved in branching, and indicate that the multiple functions of the maturase protein in group II intron branching, have been fragmented and are carried out by multiple proteins (i.e., PRP8 and YJU2) in the spliceosome, which allows for a more highly regulated process.

In summary, this manuscript presents a plethora of new information about the molecular mechanisms of group II intron splicing and, in combination with previously reported group II structures, allows a complete structural and mechanistic view throughout the group II intron splicing lifecycle. Furthermore, the structural analyses presented in this manuscript provide for the first time molecular insights into the sequential structural rearrangements that are required for the branching reaction during group II splicing, as well as and for the selection of the splice sites for subsequent exon ligation. They also elucidate additional, previously unknown molecular recognition mechanisms and functionally significant structural dynamics that are evolutionarily

conserved between the group II intron splicing machinery and the spliceosome, thereby strengthening the conclusion that the spliceosome has evolved from pre-mRNA group II introns.

We are deeply grateful for this thorough and thoughtful summary of the key findings in our manuscript and for recognizing the significance of our work to the broad RNA splicing community.

Referee #2:

The manuscript, “Structural insights into intron catalysis and dynamics during splicing”, by Xu et al. represents a significant advance in understanding the catalytic mechanism of a group II intron, particularly the new cryo-EM structure of an intron maturase complex captured just prior to the branching reaction. The structures of the same complex remodeled for exon ligation and a product complex mimic enable the authors to model the atomic interactions and rearrangements that accompany the splicing reaction. They describe structural parallels with the same reaction in pre-mRNA splicing by the spliceosome, which further reinforces support for a common evolutionary splicing ancestor. Beyond the group II intron research field, the work will be of broad interest to scientists interested in splicing mechanism, evolution of splicing, RNA structure, and RNA/protein interactions. I have no major concerns with the data and its interpretation. Minor concerns can be addressed by simple revision of some text and figures.

We thank referee #2 for highlighting the importance of our work and its implications for RNA and splicing researchers.

Specific critiques:

1. The abbreviations Pre-1F, Pre-2F and Post-2F are not intuitive. Why “F”? Avoiding the alphabet soup of the spliceosome complexes with names like, for example, “pre-branching state” or “pre-ligation complex” would be most helpful to readers.

The “F” refers to the forward splicing pathway in order to distinguish it from the reverse splicing pathway that is important for group II intron retrotransposition. These abbreviations are indicated in the cartoon of the splicing pathway shown in Fig. 1A to help orient the reader. In the final paragraph of the first results section, where these abbreviations are first mentioned, we have taken your suggestion and we now define them accordingly (pre-1F: pre-branching state; pre-2F: pre-ligation complex; post-2F: post-ligation complex) to avoid confusion. We have also added these abbreviations to the caption of Figure 1.

2. The authors should take care to emphasize that current group II introns and the spliceosome are thought to share a common ancestor molecule. Although that molecule likely more closely resembles an autocatalyzing intron, statements like “living fossil” confuses the evolutionary trajectory.

We thank the reviewer for raising this important point. We have now changed our statements in the introduction (first paragraph) and discussion (first paragraph of first discussion section) to make common ancestry of the group II RNP and the spliceosome clearer. Additionally, we have deleted the statement of “living fossil” in the second paragraph of the introduction.

3. While the $\Delta G86$ and $\Delta C601$ data in Fig. 2d supports the importance of both residues for branching, the argument that the G86A/C601U mutation demonstrates the importance of base pairing is overreaching. Deletion of nucleotides may have other structural effects relative to simply changing the base.

We acknowledge the reviewer's point and we have now modified the language in the first paragraph of results section 2 accordingly. For example, we removed the word “significant” when describing branching defects that arise from deletion mutations. For the G86A/C601U dual mutant, we think the characterization of “partially rescues branching” is not an overstatement, and does not conflict with your comment that other structural effects may be in play.

4. The large rearrangement of helix D6 following branching is striking. While Fig. 3 and the supplemental movies help to visualize the changes, it would also be nice to have a movie that highlights the interactions that must be broken to achieve the rearrangement. Can the authors further speculate what might drive this arrangement in relation to the correlating change in the spliceosome?

We agree, and in response, we have added a movie (Supplementary Video 1) that walks through the contacts with D6 that position it in the “up” conformation to organize the branch helix for the first step of splicing.

In terms of the spliceosome, it is more complicated as there are essential protein factors, such as helicases, that contribute to the remodeling of the otherwise rigid core during the transition between branching and exon ligation. Hence, the driving forces for spliceosomal branch helix rearrangement could differ from that of the group II intron, which relies solely on the free energies originating from RNA components, such as the scission of the 5'-SS, bpA reorganization, and release of the J5/6 backbone kink, which we have discussed within our manuscript (third results section). While we hesitate to speculate extensively on spliceosomal components that drive the rearrangements described in this manuscript, we have noted that Yju2 might facilitate the conformational change as described in the discussion.

5. All yellow labels in figures need to be changed to black or something with enough contrast to be legible.

We have now modified the yellow labels by adding a background or border to increase contrast for legibility.

6. When comparing the group II intron to spliceosome structures, it would be helpful if the position of intron nucleotides relative to the branch point adenosine could also be indicated. The spliceosome intron nucleotide numbering does not make this comparison intuitive. There are also labeling errors in Ext. Data Fig. 5.5 for both U2 snRNA and the intron. Labeling the snRNA outright will help with confusion between the second U of the intron and the snRNA.

We agree with these very helpful points and we have now indicated the numbering of the bpA and its adjacent nucleotides to allow easy interpretation of nucleotide position relative to the bpA in Ext. Data Fig. 6 (previously Ext. Data Fig. 5). The labels of Ext. Data Fig. 6b have also been corrected to show the U2 snRNA-intron components of the branch helix.

7. The resolution of Ext. Data Figs 2 and 3 needs to be improved. Currently, none of the graph labels in panels b-d are legible.

Thank you for noticing this. We have now made changes to Ext. Data Fig. 2 and 3 by separating Ext. Data Fig 2 into two figures, enlarging the graphs, and preparing the figures at a higher resolution so that the graph labels are legible.

Referee #3:

This manuscript reports cryoEM structures of a group IIC intron in complex with the associated maturase protein. Group II introns are ancestral prototypes for spliceosomes, offering a fundamental model for understanding RNA splicing and the evolutionary progression of spliceosomes from retroelements. Prior endeavours to capture Group II intron complexes that elucidate the branching pathway had proven ineffective. Additionally, the role of the maturase in facilitating branching remained elusive. Both of these aspects are of paramount importance, as branching is evolutionary conserved, and the maturase stands as a vestige of the spliceosome protein PRP8.

Here, changes in the preparative biochemistry, particularly the incubation with the maturase that promotes branching, and the replacement of Mg²⁺ with Ca²⁺ enabled stalling precursor and branching intermediates for cryoEM reconstruction.

Key findings include:

The mechanism by which the maturase favors branching instead of hydrolysis, by holding D6 in an orientation that brings the branch point adenosine (bpA) to the catalytic center.

The branching mechanism involves recognizing and positioning the bpA, and using a lysine finger (K361), which binds the 5'SS to juxtapose the nucleophile with the scissile phosphate.

The substrates-exchange mechanism, by a large-scale swinging of D6 that pulls the lariat and replaces it with the 3'SS.

Novel lines of evidence for the common ancestry of spliceosomes and group II introns, with respect to the dynamics of RNA-RNA and RNA-protein contacts during branching.

Some findings are entirely unexpected, while others elucidate mechanisms proposed decades ago. The data are of high quality, and the findings hold significance for a wide readership interested in RNA metabolism and evolutionary biology. Considering these merits, I recommend the publication of this work in Nature after addressing the specific points outlined below.

We thank referee #3 for highlighting the technical and conceptual breakthroughs presented in our manuscript and for recognizing its appeal to the scientific community.

Specific comments

1. The mechanistic comparison shows that the UP and DOWN positions of the D6 intron domain have equivalents in the branch duplex positions in B* and C complexes (Fig. 5b,c). The D6 domain is kept in the UP position by extensive interactions with D1 and the maturase domains (Fig. 2). Which spliceosome components interact with the branch duplex in B* and C complexes to stabilize the UP and DOWN positions, other than the Thumb-DBD of the maturase? An analysis might extend the insight into the evolution of spliceosomes, where the absence of D1 can be compensated by stage-specific proteins and PRP8 domains absent in the maturase.

In the spliceosome, the branch helix is stabilized in the “up” position (B*/C complex) by the first step factor Yju2, which potentially performs an analogous role as the ν - ν' interaction (D1c-D6). The branch helix is stabilized in the “down” position (C* complex) by the second step factor Prp18, seemingly in place of the π - π' interaction (D2-D6). This suggests that the RNA-RNA

interactions in group II introns may have been replaced by protein-RNA interactions in the spliceosome, allowing for further fine tuning and regulation of branch helix movement.

2. Is it conceivable that K361 of the maturase complements the catalytic site by neutralizing charges developed during the phosphoryl transfer, upon branching? Can minor local rearrangements possibly place K361 in the appropriate distances and geometry to influence catalysis?

This is an interesting idea to think about. While this may be possible, cryoEM can only capture static states and we are limited by local resolution. We therefore hesitate to draw specific conclusions about the mechanistic role of K361 and small local rearrangements that might occur during the transient chemical reaction. But this is a very interesting idea for downstream testing and analysis in the future.

3. The structure-guided mutations in the maturase have a drastic effect on lariat formation (Fig. 2f, lanes 2-4). It is important to check experimentally that the mutations do not disrupt the native fold of the maturase.

Thank you for raising this concern. Fortunately, the maturase that we use in our splicing assays also happens to be a potent and processive reverse transcriptase (RT). This activity requires proper native folding of the protein, and it provides an orthogonal catalytic assay for functional folding of the maturase. To address the reviewer's concern, we conducted primer extension assays with all the mutants, using a structured RNA template and DNA primer, as shown below. We observed that all proteins described in Fig. 2e retain robust RT activity, indicating that the mutations we introduced have not disrupted folding of the maturase.

* WT1 and WT2 are duplicate positive controls

4. Fig. 1d requires a more detailed description, possibly in the results section or the legend. As the maturase is present in lanes 3-5, the difference between the three samples is unclear. I understand the post-ligation RNP from lane 4 contains an oligonucleotide equivalent to the ligated exon. Are there any other differences? Besides, an additional band below the precursor is not indicated.

Thank you for this suggestion. You are correct that lanes 3-5 have maturase present in the reaction. The third lane (pre-1F and pre-2F) and the fourth lane (post-2F) indicate the FPLC-purified samples we used for grid preparation for cryoEM. The fifth lane, now renamed splicing products, is a control reaction, where the maturase-mediated splicing reaction is driven to completion, which helps us define the location of the lariat band on the gel. We have now included more details in the methods section and we have modified the legend to provide more clarity.

The additional bands, under the precursor, in Fig. 1d correspond to the linear, and linear+3' exon splicing intermediate, and these have now been labeled.

5. The gels from Fig. 2d,f look rather different than 1d, although both variants denote splicing gels. The linear fragment and the linear-3'exon are not indicated in 1d.

Yes, you are correct that these are both splicing gels, but there are slight differences in how they are prepared. Fig. 1d is a PAGE gel with fluorescently stained RNA. Fig. 2d,f come from larger PAGE gels with radiolabeled RNA, imaged by phosphor-imaging. To avoid confusion, we have now added a paragraph in the Methods Section describing how Fig. 1d was produced. We have also indicated the competing linear intron splicing pathway results in the corresponding bands shown in Fig. 1d (see response to comment #4 above).

6. Fig. 2a looks overcrowded in the central area. Perhaps it will be easier to discern the various elements by removing the density (or by other means). The bpA should be indicated for the sake of orientation. I suggest indicating how the detailed views from Fig. 2b,e relate to the overview from 2a.

We agree with this point. We have now modified Fig. 2a by zooming in on D6, removing the mesh, and simplifying the cartoon depiction. The bpA has also been enlarged and labeled to make it more prominent. The views relative to Fig. 2b,e-g (previously Fig. 2b,e) are also now indicated in Fig. 2a. We appreciate this feedback on enhancing the clarity of the figures.

7. Labeling the domains in the figures that show details of interacting residues, such as Figs 2b,e, 3a-c, facilitates the analysis (e.g., D6, exon, U2 snRNA, etc).

We appreciate the suggestion and we have implemented the changes to Fig. 2b,e-g (previously Fig. 2b,e) and Fig. 3a-c.

8. A figure showing clearly the contacts between D6 and the maturase contacts relative to the bpA (A632) would be helpful. The ED Fig. 8a is useful, but insufficient.

We agree, and have made changes accordingly that also address concerns by Reviewer 2. Specifically, Fig. 2a has now been modified to more clearly show the contacts between D6 and the maturase. This also complements changes made to ED Fig. 9a (previously ED Fig. 8a, see reviewer #3, comment #6 above). To provide additional context and perspective, we have created a short movie that walks through the interactions with D6 (see reviewer #2, comment #4).

9. Label the conformational states in Fig 3e. Fig. 3d might require optimization for conveying a message, as the indicated changes are hard to follow.

Thank you for raising this point. We have now rearranged and resized the labels in Fig. 3d, and we have labeled the conformational states in Fig. 3e to improve figure clarity.

10. To better appreciate the maturase location and impact on the RNA structure, I would indicate its location (possibly as a shadow) in Fig. 3f or a similar figure from the Ext. Data.

Thank you for the thoughtful suggestion. We have now included a cartoon of the maturase DBD in the secondary structure in Fig. 3f-h to indicate the role of the protein within the RNP complex.

11. A comparative view between pre-1F and the spliceosome, as in Fig. 5a bottom is required, especially since the parallels between the branching in the two splicing systems are a highlight of the paper.

We agree with this comment. We have prepared a comparative view between the pre-1F and the spliceosome C complex (Extended Data Fig. 8c), that illustrates the similarities between the catalytic cores of the splicing systems.

12. The branch helix is not labeled in Fig. 5 and Ext. Data Fig. 8. Only the U2 snRNA is indicated, leading to the potential confusion that lone U2 is the equivalent of D6.

Thank you for bringing our attention to this issue with the figures. We have now updated Fig. 5c with labels and modified the color scheme to clarify that the branch helix is composed of both the U2 snRNA and the intron RNA. We have also updated Ext. Data Fig. 9 (previously Ext. Data Fig. 8), in the same manner.

13. It is not clear enough the extent to which the post-2F state differs from previously published structures of group II introns. This should be indicated in the manuscript.

We thank the reviewer for the comment regarding the previous post-2F structure of a group II RNP (PDB: 5G2X; Qu et al, NSMB, 2016). While the two structures share the same overall organization, our structure is of a higher resolution (3.0Å versus 3.6Å). Therefore, we can unambiguously assign the catalytic metals, and define the specific bonding patterns that engage the bpA, which was not possible in the earlier structure. Hence, a detailed atomic and mechanistic comparison between the two structures is not possible and we did not include such a comparison in the manuscript.

14. The title's reference to "intron catalysis" is ambiguous, potentially obscuring whether it relates to pre-mRNA splicing or group II introns. Given the paper's emphasis on a group II intron mechanism, a more specific title would enhance clarity. The authors may wish to double-check this aspect.

We respectfully disagree with the reviewer on this point. Our manuscript includes mechanistic and comparative studies of group II intron RNPs and the spliceosome, and many of our findings are generalizable to pre-mRNA splicing and we therefore think that our study is broadly applicable

to general intron splicing. As such, we would prefer not to change the manuscript title.

Reviewer Reports on the First Revision:

Referees' comments:

Referee #2 (Remarks to the Author):

The authors have addressed my concerns. I recommend publication.

Referee #3 (Remarks to the Author):

The authors have been very responsive to my comments. They have addressed all my points and concerns thoughtfully and completely. The work is well done, and the manuscript and figures are notably improved. I have no further comments on the manuscript. I believe that a broad scientific community interested in RNA biology will appreciate the novel and unanticipated mechanistic features of the group II intron structure presented here.